# O-GlcNAcylation mediates Wnt-stimulated bone formation by rewiring aerobic glycolysis

Chengjia You [iD][1], Fangyuan Shen[1], Puying Yang[1], Jingyao Cui[1,2], Qiaoyue Ren[1], Moyu Liu[1], Yujie Hu[2], Boer Li[1,2], Ling Ye [iD][1,2✉] & Yu Shi [iD][1✉]

## Abstract

**Wnt signaling is an important target for anabolic therapies in osteoporosis. A sclerostin-neutralizing antibody (Scl-Ab), that blocks the Wnt signaling inhibitor (sclerostin), has been shown to promote bone mass in animal models and clinical studies. However, the cellular mechanisms by which Wnt signaling promotes osteogenesis remain to be further investigated. O-GlcNAcylation, a dynamic post-translational modification of proteins, controls multiple critical biological processes including transcription, translation, and cell fate determination. Here, we report that Wnt3a either induces O-GlcNAcylation rapidly via the $Ca^{2+}$-PKA-Gfat1 axis, or increases it in a Wnt-β-catenin-dependent manner following prolonged stimulation. Importantly, we find O-GlcNAcylation indispensable for osteoblastogenesis both in vivo and in vitro. Genetic ablation of O-GlcNAcylation in the osteoblast-lineage diminishes bone formation and delays bone fracture healing in response to Wnt stimulation in vivo. Mechanistically, Wnt3a induces O-GlcNAcylation at Serine 174 of PDK1 to stabilize the protein, resulting in increased glycolysis and osteogenesis. These findings highlight O-GlcNAcylation as an important mechanism regulating Wnt-induced glucose metabolism and bone anabolism.**

**Keywords** Wnt; O-GlcNAcylation; Glucose Metabolism; Bone Formation; Fracture Healing
**Subject Categories** Development; Post-translational Modifications & Proteolysis; Signal Transduction

See also: S Pohl and T Schinke

## Introduction

Osteoporosis is a widespread bone disease. It reduces bone mass and raises the risk of fractures by increasing the rate of bone resorption and decreasing bone formation. Osteoblasts, which are derived from mesenchymal stem cells (MSCs) are the primary bone-forming cells responsible for generating proteins that comprise the bone matrix, as well as maintaining bone mass during bone homeostasis (Long, 2011). Glucose is a primary source of both energy and carbons for the cellular functions of osteoblasts (Lee et al 2017). Osteoblast differentiation is impaired both in vitro and in vivo by osteoprogenitor-specific deletion of glucose transporter type 1 (Glut1), the most abundant glucose transporter in osteoblasts (Wei et al, 2015). Upon uptake, glucose is first metabolized by the rate-limiting enzyme hexokinase 2 (HK2) to form glucose-6-phosphate, then further metabolized by a series of glycolytic enzymes such as phosphofructokinase (PFK), glyceraldehyde-3-phosphate dehydrogenase (GAPDH), phosphoglycerate kinase 1 (PGK1), and pyruvate kinase muscle isozyme 1/2 (PKM1/2). Eventually, glucose is converted to pyruvate, which either undergoes the tricarboxylic acid (TCA) cycle in the mitochondria or transforms to lactate in the cytoplasm by lactate dehydrogenase A (LDHA) (Lee et al, 2017). Of note, as a gatekeeper of glycolysis, pyruvate dehydrogenase kinase 1 (PDK1), suppresses pyruvate dehydrogenase (PDH) to limit the influx of pyruvate into mitochondria, promoting lactate production (Erdem et al, 2022; Takubo et al, 2013). Lactate is a major product of glucose metabolism in osteoblasts even under normal oxygen conditions (Esen and Long, 2014) which is termed aerobic glycolysis (Warburg, 1956). Emerging evidence has shown that aerobic glycolysis is essential for the differentiation of MSCs into osteoblasts and subsequent bone formation. Pharmacological inhibition of PDK1 decreases aerobic glycolysis and completely reverses HIF1α-driven bone formation in vivo (Regan et al, 2014). The knockdown of LDHA impairs the mineralization activity of MC3T3-E1 osteoblastic cells in vitro (Nian et al, 2022). Importantly, multiple bone developmental signaling pathways such as Wnt, PTH, and BMP regulate the osteogenic differentiation of MSCs, by promoting aerobic glycolysis (Esen et al, 2013; Esen et al, 2015; Lee et al, 2018; Lee and Long, 2018). Among those potent osteogenic stimulators, Wnt3a has been shown to increase glucose consumption through the mTORC2 pathway and enhance the conversion of glucose to lactate, rather than allowing it to enter the TCA cycle to facilitate osteogenesis (Esen et al, 2013; Yang et al, 2021). However, the mechanism whereby Wnt3a activates aerobic glycolysis needs to be further understood.

About 2–5% of the glucose flux into the hexosamine biosynthetic pathway (HBP) leads to the PTM of serine (Ser) and threonine (Thr) residues in proteins through O-linked β-N-

[1]State Key Laboratory of Oral Diseases and National Clinical Research Center for Oral Diseases, West China Hospital of Stomatology, Sichuan University, Chengdu, China. [2]Department of Endodontics, West China Hospital of Stomatology, Sichuan University, Chengdu, China. ✉E-mail: yeling@scu.edu.cn; yushi1105@scu.edu.cn

acetylglucosamine (O-GlcNAc) moieties derived from the end product UDP-GlcNAc (Ma et al, 2021; Marshall et al, 1991; Marshall et al, 2004). The addition and removal of O-GlcNAc on protein substrates are mediated by a single pair of enzymes, O-GlcNAc transferase (OGT) and O-GlcNAc hydrolase (OGA), respectively (Chatham et al, 2021; Yang and Qian, 2017). Levels of metabolites such as glucose drive the flux via HBP, but the overall output is tightly controlled by the first rate-limiting enzymes, glutamine fructose-6-phosphate amidotransferase 1 (GFAT1) and 2 (GFAT2), which convert fructose-6-phosphate (fruc-6-P) to glucosamine-6-phosphate (GlcN-6-P) (Marshall et al, 1991). Although GFAT1 is ubiquitous, GFAT2 is only expressed in the central nervous system (Oki et al, 1999; Ruegenberg et al, 2020). The GFATs are regulated at different levels, including the modulation of mRNA and protein expression (Chaveroux et al, 2016; Manzari et al, 2007; Moloughney et al, 2016; Sayeski et al, 1997; Wang et al, 2014), post-translational modifications (Gelinas et al, 2018; Moloughney et al, 2018; Ruegenberg et al, 2021; Zhou et al, 1998; Zibrova et al, 2017), and allosteric regulation by metabolites (Mouilleron et al, 2008; Ruegenberg et al, 2020). Importantly, defects in HBP and O-GlcNAcylation that result in abnormal protein glycosylation have been linked to tumorigenesis, diabetes, insulin resistance, and other pathological conditions, most likely due to the deregulation of GFAT1 (Buse, 2006; Wang et al, 2014; Ying et al, 2012). Emerging evidence has shown that dynamic O-GlcNAcylation is implicated in multiple tissues and serves as an imperative mediator in the regulation of numerous critical biological processes, including gene transcription, translation, metabolic reprogramming, and stem cell fate determination (Chatham et al, 2021; Yang and Qian, 2017). O-GlcNAcylation is also involved in osteogenesis but the results so far have shown some inconsistency. One study found that the overexpression of OGT, which leads to massive O-GlcNAcylation, diminished BMP2-induced mineralization in C2C12 cells (Gu et al, 2018). In contrast, O-GlcNAcylation was found to increase during osteoblast differentiation in osteoblastic MC3T3-E1 cells in another study (Nagel et al, 2013). Using the same cell line, other studies reported that abolishing O-GlcNAcylation via OGT deficiency inhibited osteogenesis (Koyama and Kamemura, 2015; Weng et al, 2021). Mechanistically, the O-GlcNAcylation of the osteogenic master regulator Runx2 promotes osteogenesis in MSCs (Kim et al, 2007; Nagel and Ball, 2014). Most recently, Zhang et al reported that ablating OGT in bone marrow MSCs impaired bone formation but promoted marrow adiposity in vivo. Further investigations demonstrated that Ser$^{32}$, Ser$^{33}$, and Ser$^{371}$ are the O-GlcNAcylation sites of Runx2 (Zhang et al, 2023). O-GlcNAcylation is therefore involved in bone formation. However, besides Runx2, osteogenesis is controlled by many other factors and pathways, and several intracellular proteins are also dynamically modified by O-GlcNAcylation. Therefore, a complete understanding of how O-GlcNAcylation regulates osteogenesis warrants further studies

It is worth noting that multiple glycolytic enzymes are O-GlcNAcylated, such as PFK1 (Yi et al, 2012), PGK1 (Nie et al, 2020), PFKFB3 (Lei et al, 2020), and GAPDH (Park et al, 2009). For instance, in response to hypoxia, O-GlcNAcylation at PFK1's Ser$^{529}$ site inhibited PFK1 activity, thereby increasing pentose phosphate oxidative flux (Yi et al, 2012). In another study, PGK1 was reversibly and dynamically modified with O-GlcNAcylation at Thr$^{255}$ and activated to enhance lactate production, which

simultaneously induced translocation into mitochondria (Nie et al, 2020). Moreover, the O-GlcNAcylation of PFKFB3 at Ser$^{172}$ has been shown to control tumor proliferation under metabolic stress (Lei et al, 2020). Thr$^{227}$, as the major GAPDH O-GlcNAcylation site, mediates the nuclear translocation of GAPDH (Park et al, 2009). These findings indicate an essential role for O-GlcNAcylation in glycolysis and metabolic reprogramming. However, the potential O-GlcNAcylation of PDK1, the critical enzyme that controls the metabolic fate of pyruvate, has not been described. Moreover, whether the dynamic regulation of O-GlcNAcylation by glycolytic enzymes, particularly PDK1, coordinates the rewiring of glucose metabolism by Wnt ligands and thus promotes bone formation, remains unclear.

In this report, we show that O-GlcNAcylation is essential for Wnt-induced osteogenesis both in vivo and in vitro. Briefly, the inhibition of O-GlcNAcylation by either chemical antagonists or small interfering RNAs (siRNAs) targeting the HBP enzymes Gfat1 and Ogt significantly abolishes Wnt-induced osteogenesis of MSCs in vitro. Notably, the depletion of Ogt in osteoblast-lineage cells in vivo dramatically diminishes bone formation and postpones bone fracture healing upon Wnt activation by the administration of sclerostin-neutralizing antibodies in vivo. Mechanistically, Wnt signaling increases O-GlcNAcylation in both β-catenin-dependent and independent manners, which subsequently stimulates aerobic glycolysis. We observed that PDK1 is O-GlcNAcylated at Ser$^{174}$, and this site-specific glycosylation improves PDK1 stabilization and aerobic glycolysis induced by Wnt3a, which in turn facilitates osteoblastogenesis. This study reveals a pivotal role of O-GlcNAcylation in bone homeostasis and indicates the potential for targeting O-GlcNAc signaling to prevent metabolic osteoporosis.

## Results

### Wnt signaling increases protein O-GlcNAcylation during osteoblastogenesis

As a potent stimulator of osteogenesis, 50 ng/ml of recombinant human Wnt3a (rhWnt3a) was first confirmed to significantly increase osteogenic gene expression in the MSC line ST2 (Fig. EV1A–C). To explore the role of Wnt signaling in affecting cellular metabolism, ST2 cells treated with rhWnt3a or vehicle for 2 days were assessed for metabolites through liquid chromatography-mass spectrometry (Fig. 1A). Treatment with rhWnt3a increased multiple metabolites, among which the end-product of HBP, UDP-GlcNAc, was significantly enriched (Fig. 1B). To monitor the dynamic changes of O-GlcNAcylation during osteoblastogenesis, ST2 cells were treated with either rhWnt3a or vehicle for 72 h. O-GlcNAc level was assessed through immunoblotting, which showed that overall O-GlcNAcylation was markedly upregulated in the presence of rhWnt3a (Fig. 1C). The first rate-limiting enzyme of HBP, GFAT1, was rapidly induced by rhWnt3a as early as 6 h, and increased continuously thereafter (Fig. 1D). OGT and OGA levels were both increased markedly after 24 h treatment, which suggested precise negative regulation by OGA under conditions of high O-GlcNAcylation (Fig. 1E). Therefore, rhWnt3a increased the level of HBP enzymes and the accumulation of UDP-GlcNAc, consequently facilitating overall O-GlcNAcylation. In vivo, Wnt signaling can be activated by

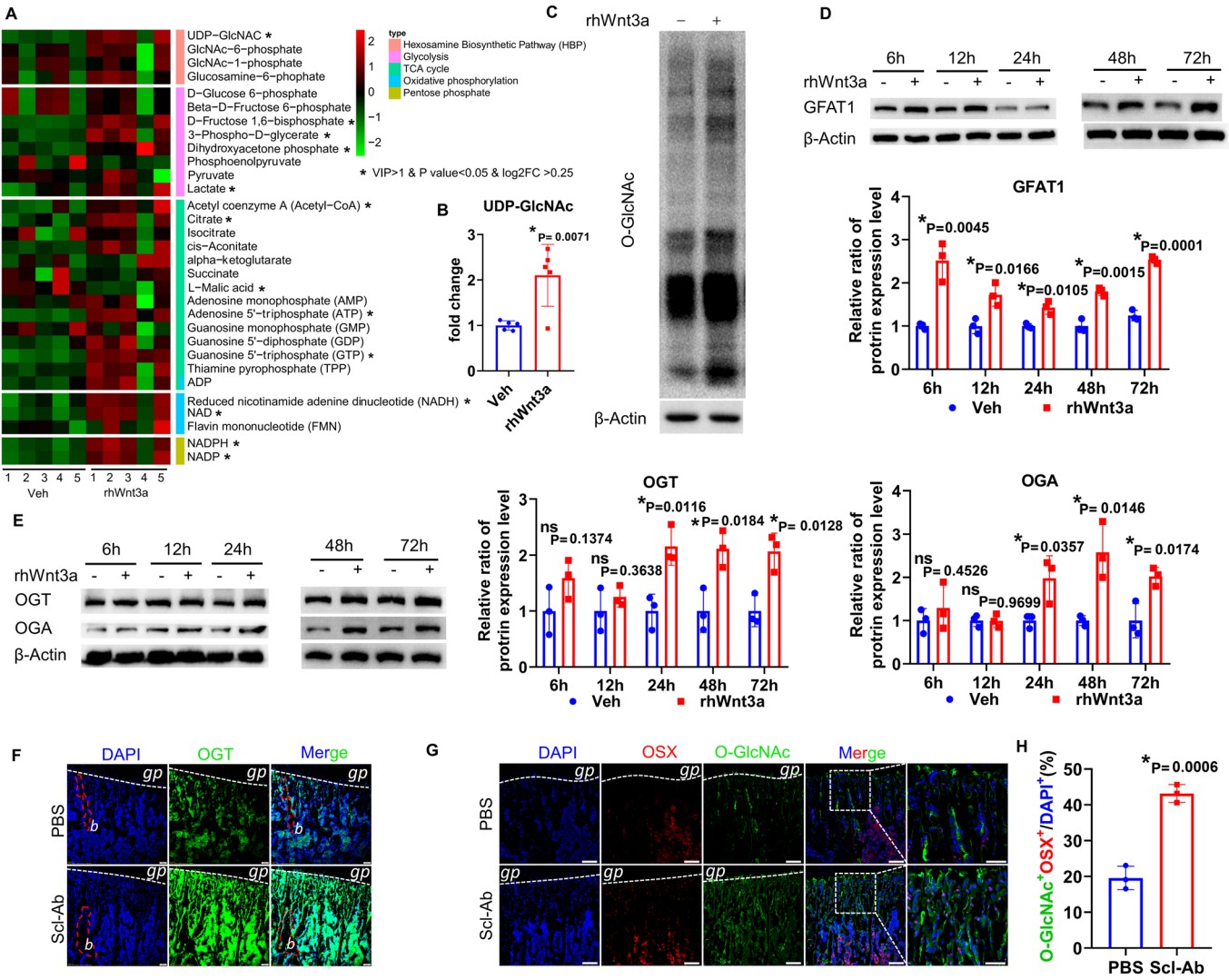

**Figure 1. Wnt3a increases protein O-GlcNAcylation.**

(A) Metabolites were measured by LC-MS after rhWnt3a treatment for 2 days. (B) UDP-GlcNAc levels were analyzed based on LC-MS. $n = 5$. (C) ST2 cells were treated with rhWnt3a for 72 h and O-GlcNAc levels were detected by Western blotting. Representative Western blots of GFAT1 (D), OGT, and OGA (E) in ST2 cells are shown. The densitometric analysis of individual protein expression was normalized to β-Actin. Quantitative analysis was performed setting vehicle treatment at each time point to standard 1, respectively. The quantifications of Western blots are presented next to the images. ns, not significant. (F, G) Representative confocal images from frozen tibia sections of mice injected with PBS as vehicle or Scl-Ab. Immunofluorescence staining of OGT (F), OSX, and O-GlcNAc (G) is performed to show their expressions in the metaphysis. White dashed lines demarcate the boundary of the growth plate. Red dashed lines demarcate the trabecular bone. gp, growth plate; b, a representative trabecular bone. (F) Scale bar, 50 μm. (G) Scale bar, 100 μm. The boxed area in (G) is shown at a high magnification to the right, scale bar, 50 μm. (H) The ratio of O-GlcNAc$^+$OSX$^+$ over DAPI$^+$ (stained nuclei of all cells) is quantified in the primary spongiosa region extending 300 μm from the growth plate and spanning the width of the bone flanked by the periosteum. (A, B, D, E, H) Each dot represented one biological replicate. Error bars: mean ± SD. $^*p < 0.05$ (two-tailed Student's t-test). Source data are available online for this figure.

sclerostin-antibody (Scl-Ab) administration, which has been extensively studied to improve bone formation by neutralizing the Wnt inhibitor sclerostin (Ominsky et al, 2017). In our study, 21-day-old wild-type mice were injected with Scl-Ab or vehicle (phosphate buffer saline, PBS) for another 4 weeks, at 4-day intervals. Their tibias were then collected and immune-stained with OGT, O-GlcNAc, and OSX antibodies. Sclerostin neutralization notably increased OGT expression in the primary spongiosa, especially in the trabecular bone (Fig. 1F). It also raised the ratio of O-GlcNAc$^+$OSX$^+$ cells (Fig. 1G,H), which indicated that, aside

from promoting osteogenesis, Wnt signaling also activated O-GlcNAcylation in osteoblasts in vivo. Together, these results suggest that, during osteogenesis of MSCs and bone formation, Wnt signaling positively regulates O-GlcNAc glycosylation.

## Inhibition of O-GlcNAcylation impedes Wnt3a-induced osteoblastogenesis in vitro

Next, we aimed to clarify the potential role of O-GlcNAcylation in osteogenesis induced by rhWnt3a. ST2 cells were pre-treated with

either the OGT inhibitor OSMI, the GFAT1 antagonist Diazooxonorleucine (DON), or the OGA inhibitor Thiamet G (TMG) for 6 h before rhWnt3a administration for 2 additional days. Data showed O-GlcNAcylation was decreased by OSMI and DON (Fig. EV1D,E), but increased by TMG (Fig. EV1E). As expected, rhWnt3a treatment elevated osteogenesis, as indicated by osteogenic-specific staining and gene expression. Notably, ALP (Fig. 2A,D) and Alizarin Red S staining (Fig. 2B,E) showed lower signal levels in response to OSMI (Fig. 2A,B) or DON treatment (Fig. 2D,E), but higher signal levels in response to TMG administration (Fig. 2G,H) in the presence of rhWnt3a. Our qPCR evaluation also demonstrated a significant decrease in Alpl and Bglap gene expression when O-GlcNAcylation was inhibited by OSMI and DON (Fig. 2C,F). Whereas gene expressions of Alpl and Bglap were further increased by TMG pre-treatment when O-GlcNAcylation was elevated (Fig. 2I). To confirm this finding, we knocked down either OGT or GFAT1 with their specific siRNAs. The qPCR data indicated the knock-down efficiency of each siRNA (Fig. 2K,M). Consistent with the previous findings, less OGT and GFAT1 lowered rhWnt3a-induced osteogenesis as indicated by ALP staining (Fig. 2J), Alizarin Red S staining (Fig. 2L), and bglap expression (Fig. 2K,M). In vitro experiments showed that low levels of O-GlcNAcylation inhibited osteogenic differentiation under basal conditions (without rhWnt3a) (Fig. 2A–F,J–M). We speculated that this might be due to attenuated endogenous Wnt signaling. Therefore, we used IWPL6 to inhibit the Wnt signaling pathway in ST2 cells and found that the level of O-GlcNAc decreased with increasing concentration (Fig. EV1F). These data demonstrated that O-GlcNAcylation is indispensable in rhWnt3a-induced osteogenesis in vitro.

## Deletion of O-GlcNAcylation in the osteoblast-lineage diminishes Wnt-induced bone formation in vivo

To address the function of O-GlcNAcylation in Wnt-stimulated bone formation, we specifically deleted OGT from osteoblast-lineage cells by crossing Ogt$^{flox/+}$ female mice with Osx-Cre (also known as Sp7-tTA; tetO-EGFP/Cre) male mice to generate Osx-Cre; Ogt$^{flox/Y}$ (OgtCKO) male mice. Males with the Osx-Cre genotype from the same litter were used as the controls (Ctrl). According to the TetO-off feature of the Osx-Cre transgenic mice line (Rodda and McMahon, 2006), which possessed Cre activity only after doxycycline was removed, we deleted OGT in osteoblast-lineage cells when the mice were 21-day old by removing doxycycline from their drinking water (Fig. 3A). To test whether Wnt signaling-increased bone formation was mediated by O-GlcNAcylation in vivo, we intraperitoneally injected with Scl-Ab or vehicle (PBS) to the Ctrl and OgtCKO mice at 4-day intervals beginning when they were 21-day old and harvested their bones 4 weeks later (Fig. 3A). First, we validated the knockout efficiency in OgtCKO mice, and Western blotting results showed a reduction in the gray level of OGT protein in the femoral diaphysis and tibial diaphysis of OgtCKO mice (Fig. EV2A). Three-dimensional reconstructions showed significantly less bone mass in the femurs of the OgtCKO mice compared with the WT ones, with or without Wnt activation (Fig. 3B). A micro-computed tomography (CT) analysis was performed to investigate trabecular parameters such as bone volume over tissue volume (BV/TV), trabecular bone number (Tb. N.), trabecular thickness (Tb. Th.), and trabecular space (Tb. Sp.) (Fig. 3C–F; Appendix Table S1). It was noteworthy that, compared with the Ctrl

mice, the BV/TV and Tb.N. were decreased, while, Tb.Sp. was increased in the OgtCKO mice. Consistently, histomorphometric analysis by Hematoxylin and Eosin (H&E) and Masson staining showed fewer trabeculae in the OgtCKO mice compared with the Ctrl ones (Figs. 3G,H and EV2B). As expected, the mice that were injected with Scl-Ab had higher bone mass as indicated by higher BV/TV, more Tb. N. and thicker Tb. Th. in Ctrl mice (Fig. 3C–E). Notably, the cancellous bone mass was still increased by Scl-Ab in the OgtCKO mice. Similarly, double labeling of bone trabeculae showed that Scl-Ab promoted the rate of mineral apposition in OgtCKO mice (Fig. EV2C,D), indicating that the trabecular bone increase by Wnt was not completely dependent on Ogt. To further investigate the bone mass in cortical bone, a three-dimensional volumetric micro-CT reconstruction was performed (Fig. 3I). This analysis showed the increases in cortical thickness (Ct. Th.), total area (Tt. Ar.), cortical area (Ct. Ar.), and cortical area over total area (Ct. Ar./Tt. Ar.) caused by sclerostin neutralization were completely blunted in OgtCKO mice (Fig. 3J–M; Appendix Table S2). We next assessed dynamic bone formation by double-labeling experiments, via staining with Calcein and Alizarin Red (Fig. 3N). The gaps between the two labels were much thinner in the OgtCKO mice. When injected with Scl-Ab, the double labels increased markedly; however, MAR remained significantly lower in OgtCKO mice even with sclerostin neutralization (Fig. 3O). Our data strongly indicated the anabolic effect of Wnt in the cortical bone mass was dependent upon protein O-GlcNAcylation.

In previous studies, OGT inhibition has been reported to repress osteoclastogenesis during the early stage (Li et al, 2022). To assess whether the decrease in bone volume caused by the removal of OGT in osteoblast-lineage cells was due to decreased bone resorption, we also performed TRAP staining on tibia sections from both Ctrl and OgtCKO mice, with and without Sclerostin neutralization. An identical level of bone resorption was observed in the tibia of Ctrl and OgtCKO mice with or without Scl-Ab (Fig. EV2E,F). Taken together, these data clearly showed that the deletion of OGT in osteoblast-lineage cells caused the inhibition of O-GlcNAcylation, which partially abolished Wnt-stimulated bone formation.

## Deletion of O-GlcNAcylation in the osteoblast-lineage delays bone fracture healing

Bone fracture healing in long bones was previously considered to follow the process of endochondral ossification in adult mice. The neutralization of Sclerostin using specific antibodies has been reported to activate Wnt signals and accelerate bone defect repair (Liu et al, 2016; Yee et al, 2016). Since O-GlcNAcylation was indispensable for cortical bone homeostasis, thus, we next examined whether Ogt was required for Wnt-stimulated bone fracture healing. The doxycycline-containing drinking water was removed to knock down OGT from the osteoblast-lineage cells at 8 weeks of age. Concurrently, Scl-Ab was injected into either Ctrl or OgtCKO mice at 4-day intervals following the fracture-induction surgery (Fig. 4A). Calluses were measured 3 weeks after the fracture procedures by micro-CT analysis and visualized by three-dimension reconstruction (Fig. 4B), and quantified by bone mineral density (BMD) and bone volume over tissue volume (BV/TV) (Fig. 4C). We observed smaller mineralized calluses in the OgtCKO mice compared with the Ctrl, as reflected by lower BMD (Fig. 4C,D) and BV/TV (Fig. 4C,E). Importantly, regardless of the presence or absence of Scl-Ab, OgtCKO mice showed similar changes in BMD

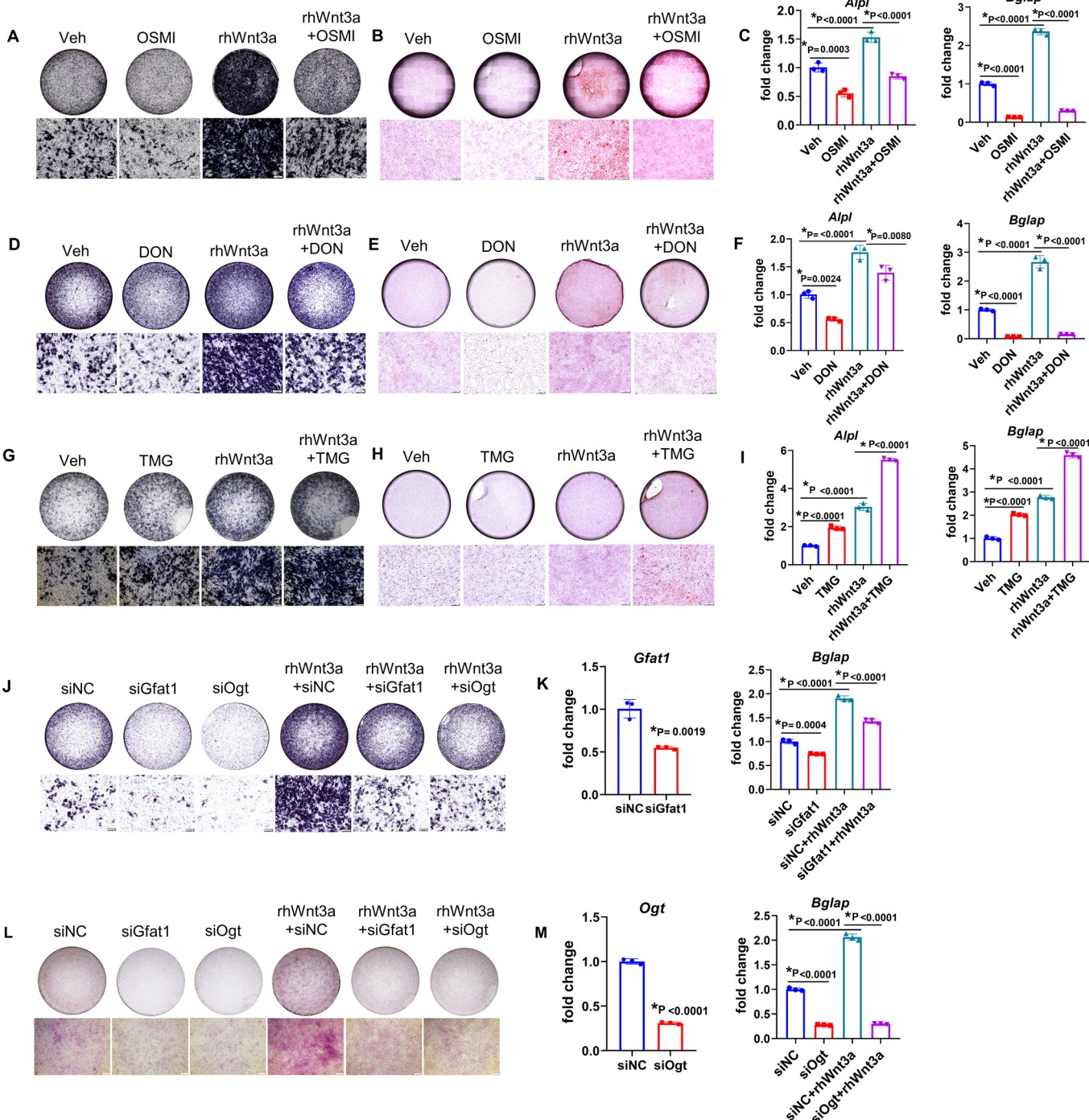

**Figure 2. O-GlcNAcylation mediates Wnt-induces osteogenesis in vitro.**

(A–I) ST2 cells were pre-treated with OSMI (A–C), DON (D–F), or TMG (G–I) before rhWnt3a treatment. (A, D, G) ALP staining and (B, E, H) Alizarin Red S staining were performed after rhWnt3a administration for 3 and 14 days, respectively. High magnifications are shown below. Scale bar = 200 μm. (C, F, I) qPCR for Alp and Bglap was performed after rhWnt3a treatment for 3 days. (J–M) Gfat1, or Ogt were knocked down before rhWnt3a treatment. (J) ALP staining, and (L) Alizarin Red S staining were performed. (K, M) qPCR for Gfat1, Ogt, and Bglap were performed rhWnt3a administration for 3 days. (C, F, I, K, M) Dots represented biological replicates. Error bars: mean ± SD. *$p < 0.05$; $n = 3$ (two-tailed Student's t-test, one-way ANOVA followed by Tukey's multiple comparisons test). Source data are available online for this figure.

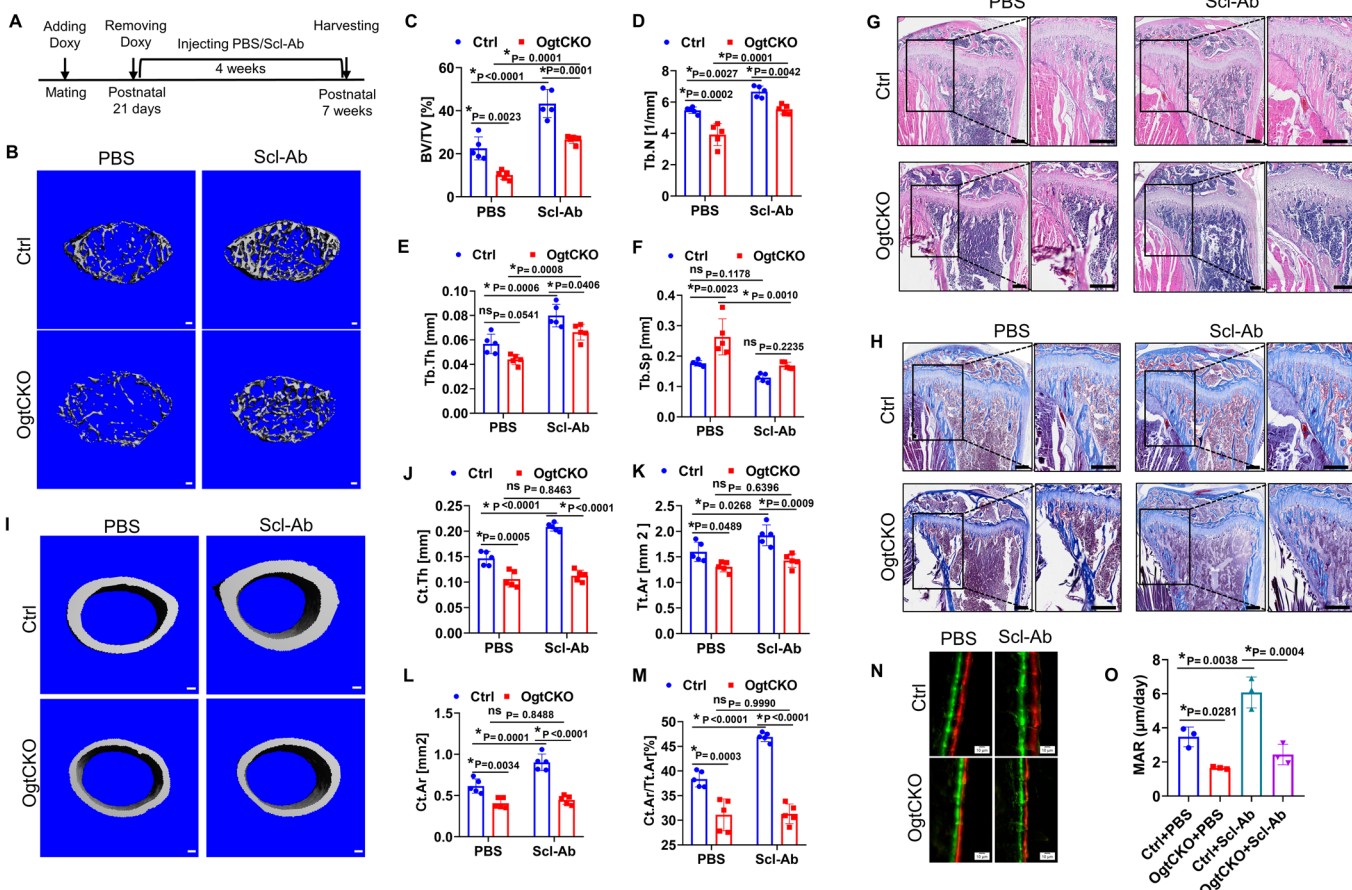

**Figure 3. Deletion of O-GlcNAcylation in osteoblast-lineage cells diminishes Wnt-induced bone formation.**

(A) A schematic of the experimental design. Representative 3D reconstruction images (B) and quantification from micro-CT scans (C–F) of the distal metaphysis of the femurs in the Ctrl or OgtCKO mice with PBS or Scl-Ab injections. $n = 5$ for each group, biological replicates. Scale bar, 100 μm. Representative H&E images (G) and Masson images (H) of the proximal tibia of the indicated genotyped mice with or without Scl-Ab injection. Boxed areas in (G, H) are shown at a high magnification to the right. Scale bar, 200 μm. Representative 3D reconstruction images (I) and quantification from micro-CT scans (J–M) of diaphyseal cortical bone of the femurs of control or OgtCKO mice with PBS or Scl-Ab injections. Scale bar, 100 μm. (N) Representative images of calcein-alizarin red double labeling in the cortical bone of femurs. Scale bar, 10 μm. (O) Quantification of Mineral Apposition Rate (MAR) is shown. (C–F, J–M, O) each dot represents one animal. Error bars: mean ± SD. *$p < 0.05$, ANOVA followed by Tukey's multiple comparisons test. Interaction $p$ values are <0.05 for treatment and genotype in (J, L, M). Source data are available online for this figure.

and BV/TV to Ctrl mice. (Fig. 4C–E). H&E (Fig. 4F) and Masson staining (Fig. 4G) revealed a decrease in cartilage (Fig. 4H) and an increase in the percentage of bone in the callus (Fig. 4I) in OgtCKO mice treated with Scl-Ab compared to Ctrl mice. The above data suggest that Wnt-mediated fracture healing is mediated by OGT, but not exclusively dependent upon OGT.

## Lack of O-GlcNAcylation diminishes Wnt3a-promoted aerobic glycolysis

To clarify how O-GlcNAcylation regulates osteogenesis, we examined the mRNA profiles of ST2 cells by performing RNA sequencing when the ST2 cells were treated with vehicle, rhWnt3a, or rhWnt3a combined with OSMI, for 3 days. There were 802 genes that responded to both rhWnt3a treatment and the addition of OSMI (Fig. 5A). The KEGG data showed that besides ossification, the glycolysis pathway was enriched in these genes (Fig. 5B). Furthermore, increased levels of expression of mRNAs for

glycolytic enzymes such as Pdk1, Eno1, Ldha, and Pgk1 were all abolished by the addition of OSMI (Fig. 5C). Importantly, Wnt ligands such as Wnt3a (Esen et al, 2013) and Wnt7b (Chen et al, 2019), have been reported to alter glucose metabolism and regulate bone formation. Data in this study also confirmed loss of Hk2 reduced osteogenic gene expression and blockade of glucose metabolism by 2-Deoxy-D-glucose (2-DG) diminished rhWnt3a-induced osteogenesis in vitro (Fig. EV3A,B). Therefore, we investigated whether O-GlcNAcylation mediated Wnt3a-induced osteoblastogenesis by regulating aerobic glycolysis. To do this, we first pretreated ST2 cells with OSMI for 24 h before the administration of rhWnt3a for 48 additional hours. Glucose uptake and lactate production were then assessed, and we found a significant increase in both, in response to rhWnt3a, as we expected. It is worth noting that the inhibition of O-GlcNAcylation by OSMI significantly decreased glucose uptake (Fig. 5D) and lactate production (Fig. 5E) both with and without the administration of rhWnt3a. Seahorse assays detected a dramatic

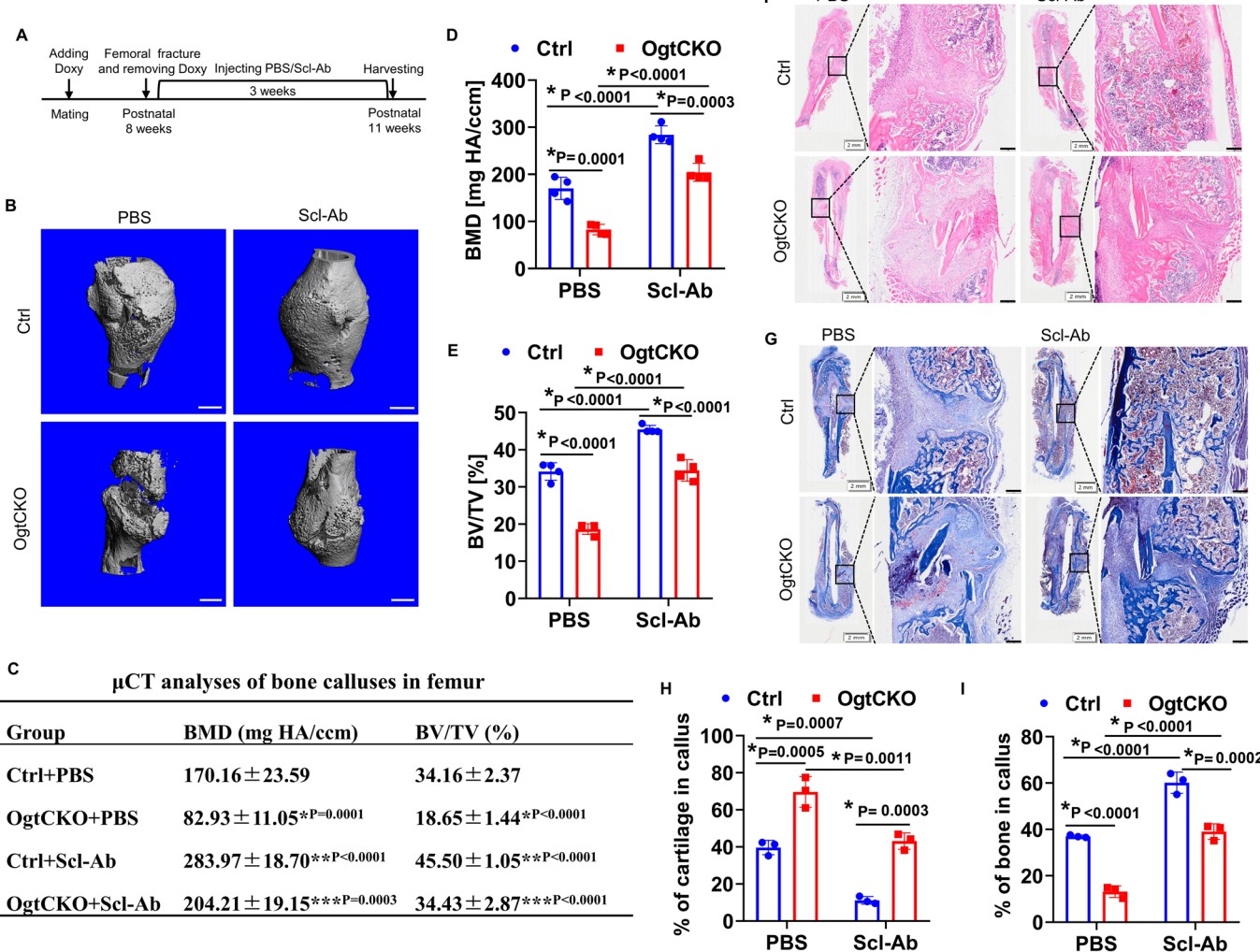

**Figure 4. Loss of O-GlcNAcylation in osteoblast-lineage cells impairs bone fracture healing.**

(A) A schematic of the experimental design. (B) Representative 3D reconstruction images. Scale bar, 1 mm. (C) Micro-CT analysis of the femoral callus at postfracture day 21. Data are shown as mean ± SD, n = 4, biological replicates. BMD = bone mineral density; BV/TV = bone volume over tissue volume. Data was acquired from slices of the whole femoral callus. *p, OgCKO+PBS versus Ctrl+PBS; **p, Ctrl+Scl-Ab versus Ctrl+PBS; ***p, OgCKO+Scl-Ab versus Ctrl+Scl-Ab, two-way ANOVA followed by Tukey's multiple comparisons test. (D, E) Quantification from (C). Error bars: mean ± SD. *p < 0.05, two-way ANOVA followed by Tukey's multiple comparisons test. n = 4 for each group, each dot represented one animal. (F, G) Representative images of H&E (F) and Masson (G)-stained sections of the medial sections through the fractured femur at day 21, scale bar, 2 mm. The boxed area in (F, G) is shown at high magnification to the right, scale bar, 200 μm. (H, I) The relative cartilaginous area and newly formed bone in the callus were measured according to the Masson staining. *p < 0.05, two-way ANOVA followed by Tukey's multiple comparisons test, n = 3, biological replicates. Source data are available online for this figure.

increase in the extracellular acidification rate (ECAR) (Fig. 5F) and identical oxygen consumption rate (OCR) (Fig. 5G) in the presence of rhWnt3a; however, the increase of ECAR was suppressed with the addition of OSMI. Thus, the decrease of O-GlcNAcylation significantly abolished Wnt-induced aerobic glycolysis. Notably, OSMI also reduced maximal respiration even in the absence of rhWnt3a, which suggested that O-GlcNAcylation also plays a critical role in maintaining mitochondrial respiration in osteoblast-lineage cells (Fig. 5G). Then, we detected the expression levels of key glycolytic enzymes in response to rhWnt3a stimulation and dynamic alteration of O-GlcNAcylation. Consistently, 48 h of rhWnt3a treatment significantly increased the protein levels of

HK2, PFKFB3, PFK1, LDHA, and PDK1. Notably, the administration of OSMI or TMG dramatically decreased and increased these protein levels, respectively, in the presence of rhWnt3a (Fig. 5H). Similarly, knocking down OGT using siRNA abolished the rhWnt3a-stimulated increases in glycolytic protein levels, while the knockdown of OGA mildly further increased the expression levels of proteins such as HK2 (Fig. 5I). We also observed that the inhibition of GFAT1 by its antagonist DON also diminished rhWnt3a-induced glycolytic enzyme levels (Fig. 5J). These results demonstrated that O-GlcNAcylation mediated Wnt-induced osteogenesis through the upregulation of aerobic glycolysis and the increase of glycolytic enzyme protein levels.

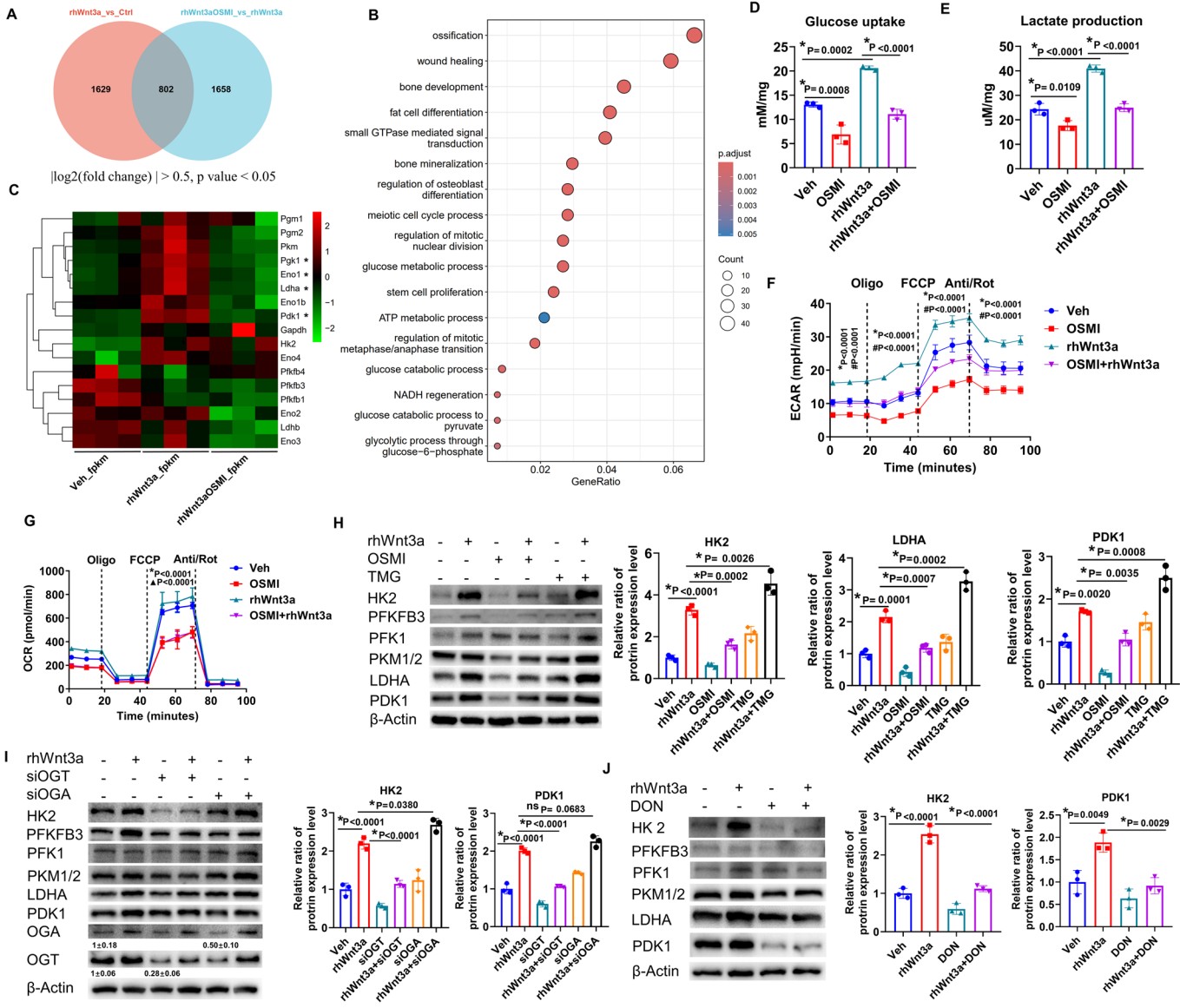

**Figure 5. O-GlcNAcylation is indispensable for Wnt3a-increased aerobic glycolysis.**

(A) Venn diagram showing that 802 genes that were altered in response to rhWnt3a and OSMI (Benjamini & Hochberg method). (B) Gene ontology (GO) analysis of the genes listed in (A) (Hypergeometric test). (C) Heatmap of the representative glycolytic genes in response to the indicated treatments, n = 3, biological replicates. (D, E) Glucose uptake (D), and lactate production (E) were performed in the indicated groups. (F, G) ECAR (F), and OCR (G) were exhibited to represent the capacity of aerobic glycolysis and OXPHOS, respectively. (H) Representative Western blots are shown to reflect the expressions of glycolytic enzymes in response to rhWnt3a, OSMI, or both. Quantifications from densitometric analysis of HK2, LDHA, and PDK1 are shown on the right. (I) ST2 cells were pre-treated with siRNA against either OGT or OGA, and then treated with rhWnt3a for 2 days. Quantitative analysis of siRNA knockdown efficiency was performed, data are shown as mean ± SD, n = 3, biological replicates. Representative Western blots are shown and the quantifications from densitometric analysis of HK2 and PDK1 are shown on the right. (J) Representative Western blots are shown to demonstrate the effect of DON on Wnt-induced glycolytic enzyme expression. The quantifications of HK2 and PDK1 are presented. (D, E, H–J) *p < 0.05, n = 3; dots represented biological replicates, one-way ANOVA followed by Tukey's multiple comparisons test. (F, G) # represents the comparison between rhWnt3a and vehicle treatment; ▲ represents the comparison between OSMI and vehicle treatment; * represents the comparison between rhWnt3a and rhWnt3a+OSMI treatment. Error bars: mean ± SD. #, ▲, *p < 0.05, two-way ANOVA followed by Tukey's multiple comparisons test, n = 3, biological replicates. Source data are available online for this figure.

## Canonical Wnt signaling mediates the induction of O-GlcNAcylation and improves aerobic glycolysis

Next, we planned to understand how Wnt3a orchestrated O-GlcNAcylation in glycolysis. A previous study showed that Wnt3a rapidly stimulates aerobic glycolysis in osteoblasts within

6 h via the mTORC2-AKT axis, but that β-catenin-dependent Wnt signaling does not have the same effect (Esen et al, 2013). However, β-catenin has also been reported to be involved in glucose metabolism during tumorigenesis, which suggests that canonical Wnt signaling is crucial to glucose metabolism in tumor cells at least, if not in all cells (Nie et al, 2022; Pate et al, 2014; Yang et al,

2021). Thus, we asked whether the prolonged administration of rhWnt3a might regulate aerobic glycolysis through β-catenin-dependent Wnt signaling during osteogenesis. To assess this, we first treated ST2 cells for 72 h with lithium chloride (LiCl), a well-studied canonical Wnt stimulator that inhibits glycogen synthetase kinase-3β and consequently stabilizes cytosolic β-catenin. We found that glucose uptake (Fig. 6A) and lactate production (Fig. 6B) were both increased. The seahorse assay indicated cells manifested increased ECAR, but relatively identical OCR in response to LiCl (Fig. 6C,D). Glycolytic proteins such as HK2, PFK1, PDK1, and PKM1/2 were also significantly upregulated in presence of LiCl (Fig. 6E). To confirm this finding, we pre-treated ST2 cells with the β-catenin-specific inhibitor XAV939, which targeted the poly-ADP-ribosyltransferases tankyrase 1&2 to destabilize β-catenin, for 12 h, followed by the administration of rhWnt3a for an additional 72 h, after which glucose uptake (Fig. 6F) and lactate production (Fig. 6G) were measured. XAV939 barely had any effect on these assays in the absence of rhWnt3a; however, the induction of glucose uptake and lactate production by rhWnt3a were abolished by the addition of XAV939. Our seahorse experiments also showed that Wnt-induced extracellular acidification was mediated by β-catenin (Fig. 6H). Consistently, rhWnt3a did not alter the OCR. However, XAV939 decreased the basal respiration significantly regardless of rhWnt3a addition which suggested a potential role of β-catenin in mitochondrial respiration independent of Wnt ligand (Fig. 6I). Glycolytic protein expression levels such as HK2 and PDK1 increased upon rhWnt3a treatment were reduced by the inhibition of β-catenin (Fig. 6J). Two sets of siRNAs were used to knock down β-catenin, which blocked the induction of glycolytic protein expression, with or without the administration of rhWnt3a for 48 h (Fig. 6K). This indicated that β-catenin-Wnt signaling significantly regulated aerobic glycolysis following prolonged treatment of Wnt3a.

Notably, we found the O-GlcNAcylation and GFAT1 levels increased markedly in the presence of LiCl (Fig. 6L,M), while the induction O-GlcNAcylation by rhWnt3a was blunted by XAV939 (Fig. 6N). The knockdown of β-catenin not only attenuated O-GlcNAcylation levels (Fig. 6O) but also decreased the upregulation of GFAT1 and OGT protein expression induced by rhWnt3a (Fig. 6P). This confirmed that canonical Wnt signaling is critical for Wnt-induced O-GlcNAcylation. Combined with our previous finding, this shows that the prolonged administration of rhWnt3 stimulates aerobic glycolysis through O-GlcNAcylation in a β-catenin-dependent manner.

## Wnt rapidly increased O-GlcNAcylation via the Ca²⁺-PKA-GFAT1 axis

Owing to the quick changes in GFAT1 protein expression in response to rhWnt3a within 6 h (Fig. 1D) and the fact that no significant alterations took place in the short stimulation of LiCl (6 or 12 h) (Figs. 7A and EV4A), we suspected that Wnt signaling could rapidly alter GFAT1 protein level in a β-catenin-independent manner. GFAT1 is post-translationally regulated by cAMP-dependent protein kinase (PKA) (Zibrova et al, 2017), Mammalian target of rapamycin complex 2 (mTORC2) (Moloughney et al, 2018), and adenosine 5′ monophosphate-activated protein kinase (AMPK) (Eguchi et al, 2009; Zibrova et al, 2017). Therefore, we first validated the protein expression of the PKA direct target phospho-

CREB at Ser$^{133}$, the phospho-AMPK at Thr$^{172}$, and the mTORC2 direct target phospho-AKT at Ser$^{473}$, in the presence of rhWnt3a from 10 to 60 min. Among them, the phospho-CREB (Ser$^{133}$) responded quickly at 10 min (Figs. 7B and EV4B). Next, to investigate whether Wnt-induced glycolysis and O-GlcNAcylation required PKA, we pretreated ST2 cells with H89, a specific inhibitor of PKA, followed by 6 h of rhWnt3a treatment. The inhibition of PKA significantly blunted Wnt-mediated glycolytic protein expression (Figs. 7C and EV4C), GFAT1, and O-GlcNAcylation levels (Figs. 7D and EV4D). Intracellular calcium (Ca²⁺) has also been reported to interact with PKA in many tissues (Grewal et al, 2000; Ni et al, 2011). To test the effect of rhWnt3a on Ca²⁺ influx, we treated ST2 cells with rhWnt3a for the indicated time and found that rhWnt3a dramatically increased intracellular calcium flux after 260 s of rhWnt3a administration (Fig. 7E red line). Further, the L-type Ca²⁺ channel ionophore ionomycin increased Ca²⁺ influx to a certain level as rhWnt3a did (Fig. 7E green line); whilst ionomycin further enhanced Ca²⁺ signals in addition to rhWnt3a which proved that rhWnt3a treatment indeed increased Ca²⁺ influx in the cytosol (Fig. 7E purple line). To further confirm this, protein level of calcium-calmodulin (CaM)-dependent protein kinase II (CaMKKII) as well as liver kinase B1 (LKB1) were also measured and found increased in response to rhWnt3a (Figs. 7F and EV4F). The results indicated that rhWnt3a treatment increased the non-canonical pathway through elevated calcium flux and PKA activity.

Next, to investigate whether Wnt-induced GFAT1 expression required calcium influx, we pretreated ST2 cells with STO-609, a selective inhibitor of CaMKKII, followed by 6 h of rhWnt3a treatment. We found that Wnt-induced GFAT1 protein over-expression and O-GlcNAcylation levels both declined when CaMKKII was inhibited (Figs. 7G and EV4G). Additionally, our data showed that inhibition of CaMKKII also abolished the induction of glycolytic protein expression triggered by rhWnt3a (Figs. 7H and EV4H). Besides, the osteogenesis trigged by rhWnt3a was also diminished reflected by ALP (Fig. 7I) and Alizarin Red S staining (Fig. 7J). We also observed that the phospho-CREB (Ser$^{133}$) level also declined in the presence of STO-609 which indicated a regulatory effect of calcium influx on the PKA pathway (Figs. 7K and EV4 I). This suggested that rhWnt3a treatment rapidly increases O-GlcNAcyation levels and glycolysis via the Ca²⁺-PKA-GFAT1 axis.

## O-GlcNAcylation on Ser$^{174}$ of PDK1 mediates Wnt3a-induced osteogenesis

PDK1, as a gatekeeper of aerobic glycolysis, promotes the production of lactate (Erdem et al, 2022; Takubo et al, 2013). The inhibition of PDK1 diminishes HIF1α-driven bone formation (Regan et al, 2014). Since rhWnt3a treatment significantly upregulated PDK1 protein levels, and this upregulation was diminished by the inhibition of O-GlcNAcylation (Fig. 5H,I), we decided to ask whether PDK1 was directly O-GlcNAcylated. To this end, we first infected an HA-tagged OGT adenovirus to overexpress OGT in ST2 cells. Then we performed immunoprecipitation (IP) with O-GlcNAc antibody and immunoblotted (IB) with PDK1 antibody, which showed that PDK1 possessed O-GlcNAc (Fig. 8A black arrow pointed). On the other hand, we also overexpressed Flag-tagged full-length PDK1 (Flag-PDK1) by lentivirus in the ST2 cells, then performed an IP experiment with Flag antibody and IB

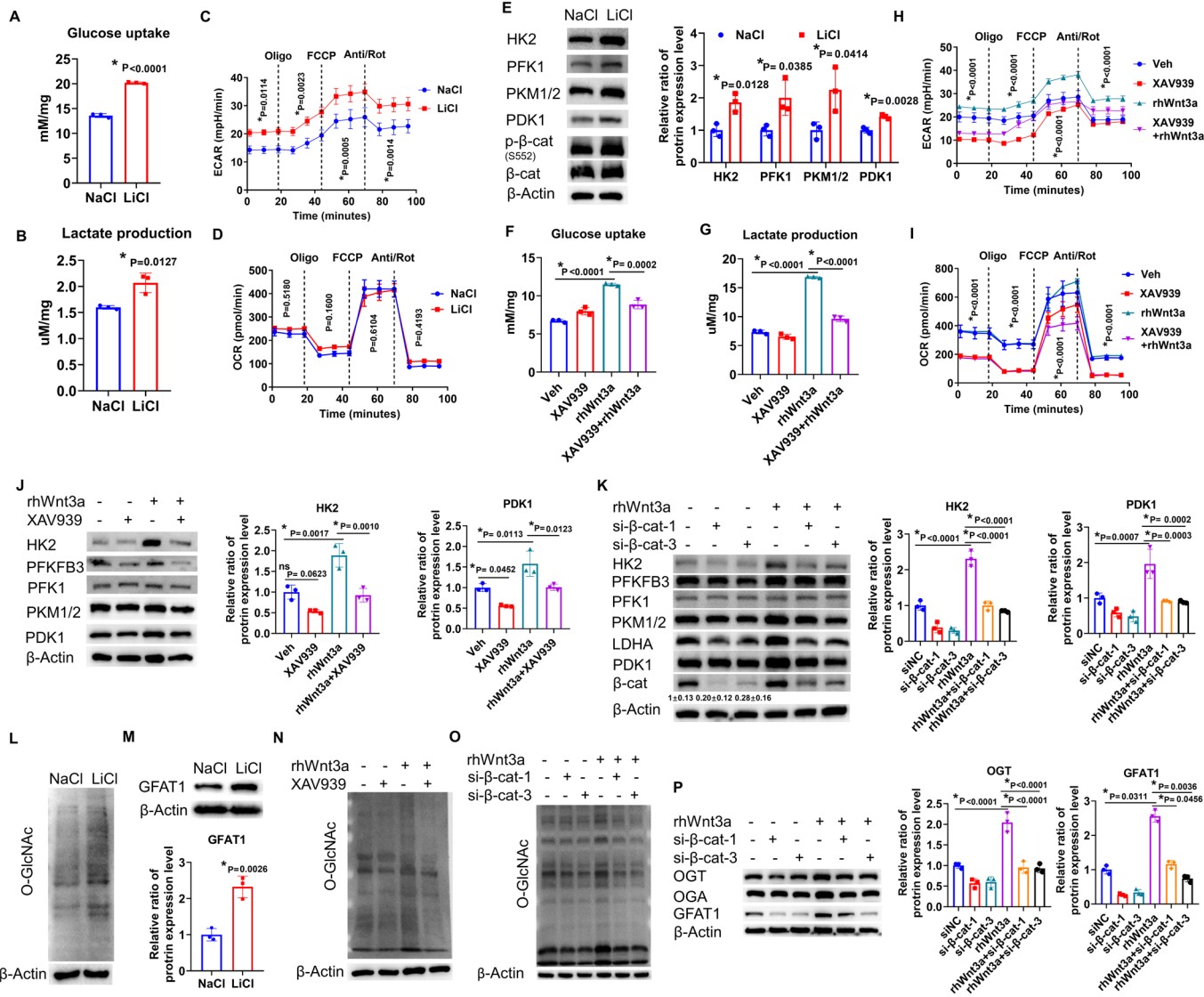

**Figure 6. Prolonged Wnt3a signaling requires β-Catenin to rewire O-GlcNAcylation and aerobic glycolysis.**

(A–D) Glucose uptake (A), Lactate production (B), ECAR (C), and OCR (D) assay of ST2 cells in response to LiCl for 72 h. *p < 0.05, two-tailed Student's t-test and two-way ANOVA followed by Sidak's multiple comparisons test, n = 3, biological replicates. (E) The representative Western blotting for glycolytic enzyme detection exposed to LiCl for 72 h, and quantification are presented. two-tailed Student's t-test, n = 3, biological replicates. (F, G) Glucose uptake (F), and Lactate production (G) of ST2 cells exposed to XAV939, rhWnt3a, or both. One-way ANOVA followed by Tukey's multiple comparisons test, n = 3, biological replicates. (H, I) ECAR (H), and OCR (I) assay of ST2 cells in response to the indicated treatments. * represents the comparison between rhWnt3a and rhWnt3a + XAV939 treatment, *p < 0.05, two-way ANOVA followed by Tukey's multiple comparisons test, n = 3, biological replicates. (J) Glycolytic enzyme expressions were examined and quantifications of HK2 and PDK1 are shown. One-way ANOVA followed by Tukey's multiple comparisons test, n = 3, biological replicates. (K) β-Catenin was knocked down before rhWnt3a treatment for 48 h. Quantitative analysis of siRNA knockdown efficiency was performed, data are shown as mean ± SD, n = 3, biological replicates. The representative Western blotting for glycolytic enzyme and quantification are presented. One-way ANOVA followed by Tukey's multiple comparisons test, n = 3, biological replicates. (L, M) Western blotting results of O-GlcNAc levels (L) and GFAT1 (M) in ST2 cells treated with LiCl for 72 h (two-tailed Student's t-test). (N) Western blotting results of O-GlcNAc levels in response to XAV939 and rhWnt3a treatment for 72 h. (O, P) ST2 cells were treated the same as those in (K), O-GlcNAc (O), OGA, OGT, and GFAT1 (P) levels were exhibited with OGT and GFAT1 quantification shown to the right. One-way ANOVA followed by Tukey's multiple comparisons test, n = 3, biological replicates. *p < 0.05. Error bars: mean ± SD. Source data are available online for this figure.

with O-GlcNAc antibody. The data showed that PDK1 possessed more O-GlcNAcylation in response to rhWnt3a treatment (Figs. 8B and EV5A). Multiple studies have proven that O-GlcNAcylation directly regulates protein stability.(Duan et al, 2018; Li et al, 2013; Li et al, 2017) To determine the potential effect of O-GlcNAcylation on the half-life of PDK1, we first overexpressed

Flag-PDK1, then blocked de novo protein synthesis using cycloheximide (CHX). We monitored the abundance of Flag-PDK1 over time with the addition of rhWnt3a, OSMI, or both. These experiments revealed that rhWnt3a treatment prolonged the half-life of Flag-PDK1, and the stabilizing effect of rhWnt3a was completely abolished by the O-GlcNAcylation inhibitor OSMI

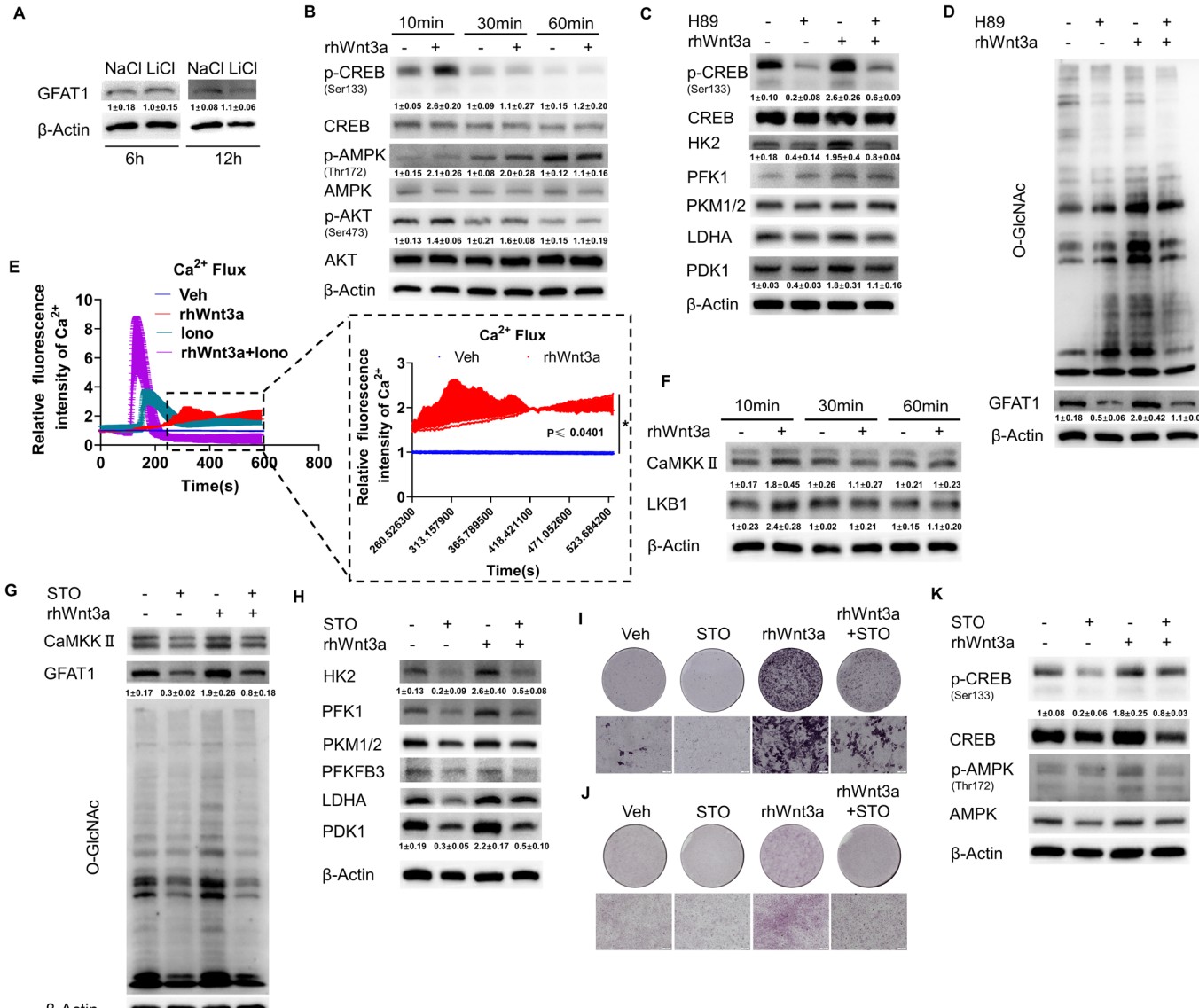

**Figure 7. Wnt rapidly increases O-GlcNAcylation via the Ca²⁺-PKA-GFAT1 axis.**

(A) GFAT1 protein levels in response to LiCl at 6 or 12 h. (B) The Western blotting analyses for pCREB (Ser$^{133}$), pAMPK (Thr$^{172}$), and pAKT (Ser$^{473}$). Quantitative analysis was performed setting vehicle treatment at each time point to standard 1, respectively. (C, D) ST2 cells were treated with rhWnt3a, H89, or both for 6 h. (C) Glycolytic enzyme expression levels were examined. (D) O-GlcNAc and GFAT1 levels were validated. (E) Calcium flux in response to rhWnt3a, ionomycin (Iono), or both in a series of time points. On the right is a statistical analysis of calcium flux after vehicle and rhWnt3a treatment for 260 s. *$p < 0.05$, two-way ANOVA followed by Tukey's multiple comparisons test, $n = 3$, biological replicates. (F) the representative Western blotting images of CaMKKII and LKB1 are shown in response to rhWnt3a with indicated time. Quantitative analysis was performed using setting vehicle treatment at each time point to standard 1, respectively. (G, H, K) ST2 cells were administrated with rhWnt3a, STO, or both for 6 h. (G) Western blotting analyses of CaMKKII, O-GlcNAc, and GFAT1. (H) The representative Western blotting image of glycolytic enzyme in ST2 cells. (I) ALP staining and (J) Alizarin Red S staining were performed after 3 and 14 days' rhWnt3a treatment respectively, in either STO or rhWnt3a. High magnificent images are shown below, scale bar, 200 μm. (K) Western blotting images of pCREB (Ser$^{133}$) and pAMPK (Thr$^{172}$) were performed in response to STO and rhWnt3a for 6 h. Data are shown as mean ± SD, $n = 3$, biological replicates. Source data are available online for this figure.

(Fig. 8C,D). As PDK1 could be directly O-GlcNacylated, we predicted the O-GlcNAc sites on PDK1 using the online tool YinOYang (http://www.cbs.dtu.dk/services/YinOYang/) with the standard criterion and found Ser$^{174}$ with high score. To test the biological functions of these sites, we performed the point mutation on Flag-tagged PDK1 and transfected Flag-PDK1 (WT) or the Flag-PDK1-S174A (S174A) into ST2 cells, making stable cell lines. Alanine substitution of serine at the Ser$^{174}$ decreased but not fully

eliminated the O-GlcNAcylation of PDK1, which suggested PDK1 possessed multiple functional O-GlcNAc sites beside Ser$^{174}$ (Figs. 8E and EV5B). Importantly, this mutation accelerated the degradation of Flag-PDK1 (Fig. 8F,G). Next, the proteasome inhibitor MG132 was used and dramatically increased the ubiquitination in S174A cells, which confirmed less O-GlcNAc modification led to instability of Flag-PDK1 (Fig. 8H). Consequently, this S174A mutation suppressed rhWnt3a-induced

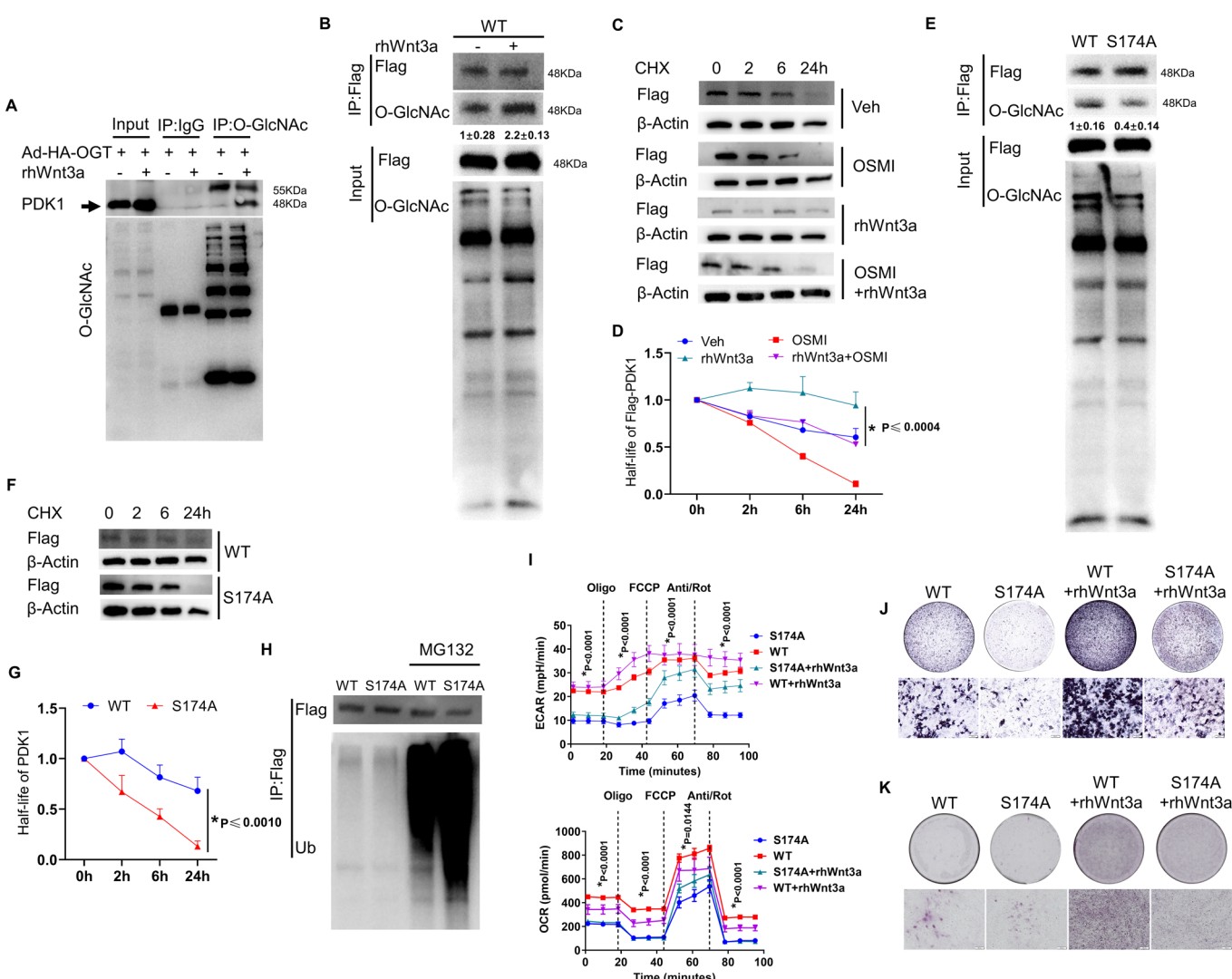

**Figure 8.** O-GlcNAcylation on the Ser[174] site stabilizes PDK1 and facilitates Wnt3a-induced osteogenesis.

(A) Immunoprecipitation of endogenous O-GlcNAc and detection with PDK1 Western-blotting. (B) Immunoprecipitation (IP) of Flag-PDK1 and detection with O-GlcNAc immunoblotting. Fold changes were quantified and shown as mean ± SD. (C, D) Effects of rhWnt3a and OSMI on Flag-PDK1 turnover, as measured by Western blotting (C). The quantifications are shown (D). *$p < 0.05$, two-way ANOVA followed by Tukey's multiple comparisons test, $n = 3$, biological replicates. (E) IP of Flag-PDK1-S174A and detection with O-GlcNAc immunoblotting. Fold changes were quantified and shown as mean ± SD. (F, G) Effects of S174A on the half-life of Flag-PDK1, as measured by Western blotting (F) and presented with quantification (G). *$p < 0.05$, two-way ANOVA followed by Sidak's multiple comparisons test, $n = 3$, biological replicates. WT, ST2 cells with Flag-PDK1; S174A, ST2 cells with Flag-PDK1-S174A mutant. (H) WT or S174A cells were treated with MG132 for 6 h and performed IP with Flag antibody, and then detected with ubiquitin antibody. (I) The seahorse assay in S174A and WT cells in response to rhWnt3a for 24 h. * represents the comparison between S174A cells with rhWnt3a treatment and WT cells with rhWnt3a treatment, *$p < 0.05$, two-way ANOVA followed by Tukey's multiple comparisons test, $n = 3$, biological replicates. (J) ALP staining and (K) Alizarin Red S staining were performed after 3 and 14 days' rhWnt3a treatment, respectively, in either WT or S174A cells. High magnificent images are shown below, scale bar, 200 μm. All data are shown as mean ± SD. Source data are available online for this figure.

extracellular acidification (Fig. 8I) and impeded osteogenesis in ST2 cells (Fig. 8J,K). Thus, rhWnt3a promoted aerobic glycolysis, partially through O-GlcNAcylation-mediated PDK1 turnover.

## Discussion

Here, we report that O-GlcNAcylation is necessary for Wnt3a-induced osteoblastogenesis in vitro, and for Wnt-activated bone formation and fracture healing in vivo. Molecularly, Wnt3a promotes O-GlcNAcylation through a Ca²⁺-PKA-Gfat1 or β-catenin-dependent manner, which in turn increases glycolysis and osteogenesis. This study not only sheds novel light on the mechanism behind Wnt signaling for regulating O-GlcNAc modification but also provides proof of principle that Wnt activated-O-GlcNAcylation may be explored for the pharmaceutical treatment of osteoporosis.

To the best of our knowledge, this is the first study to investigate the function of O-GlcNAcylation in Wnt-induced osteogenesis and bone formation. Wnt ligands are potent anabolic stimulators of

osteogenesis. Emerging evidence has shown that Sclerostin, an inhibitor of Wnt signaling, is a therapeutic target in bone loss. Its neutralization by the specific antibody Scl-Ab (also known as Romosozumab) has been reported to be an effective pharmaceutical treatment for osteoporosis (Dreyer et al, 2021; Ke et al, 2012; Lewiecki, 2014). Although a recent study (Zhang et al, 2023) indicated that loss of Ogt in Osx-positive cells in vivo led to bone defects, it is necessary to assess the function of O-GlcNAcylation in bone formation and fracture healing in response to Wnt stimulation. Therefore, in this study, rhWnt3a was used to treat ST2 cells, which are representative of MSCs in vitro, which significantly increased O-GlcNAcylation. Similarly, C3H/10T1/2 and M2-10B4 cells showed an upregulation of O-GlcNAcylation upon rhWnt3a treatment (Fig. EV1G,H), indicating that upregulation of O-GlcNAcylation of cellular total proteins by rhWnt3a may be a general phenomenon, but the extent of expression of O-GlcNAcylation was not the same in different cells due to the time, concentration and degree of cellular response to rhWnt3a treatment. Of note, whether WNT3A acts in the same way to promote osteogenesis in mice and humans has not been reported in relevant studies. Studies have shown that the WNT3A gene is strongly associated with bone fragility (Caetano da Silva et al, 2021; Velázquez-Cruz et al, 2014). Another study has shown that the WNT3A (c.152 A > G, p.K51R) mutation is found in patients with Childhood-onset primary osteoporosis and that the mutation manifests in CHO cells as a reduction in the activity of the classical Wnt pathway, supporting the role of the classical Wnt pathway in the development of the disease (Korvala et al, 2012). Notably, interference with the elevation of O-GlcNAcylation significantly induced an arrest in osteoblast differentiation in vitro, characterized by the significant transcriptional inhibition of osteogenic genes and calcium deposit. Consistent with the findings of a recent study (Zhang et al, 2023), we specifically knocked down Ogt in osteoblast-lineage cells and found a low bone mass phenotype in the OgtCKO mice. In addition to reduced bone formation and lower bone mass, the loss of O-GlcNAcylation also causes delays in bone fracture healing. In this study, the deletion of Ogt completely blocked Scl-Ab-induced cortical bone formation and regeneration after injury which indicated the anabolic effect of Wnt was dependent upon protein O-GlcNAcylation. Unlike cortical bone, the trabecular bone mass in OgtCKO despite attenuated compared to Ctrl mice, remained increasing in response to Scl-Ab administration, which suggested Wnt-stimulated bone mass in trabeculae did not fully rely on O-GlcNAcylation. The diverse response to Wnt stimulation between cancellous and cortical bone needed to be further studied. Similar to cancellous bone, Scl-Ab still partially promotes fracture healing in OgtCKO mice, and it is speculated that Wnt signaling may regulate other cytokines or pathways in addition to O-GlcNAc levels to promote fracture healing. It has been shown that BMP-2 is upregulated after 2 weeks of fracture treatment in Scl-Ab-treated rats (Feng et al, 2015), suggesting that the upregulation of the BMP signaling pathway by the Wnt signaling pathway synergistically promotes the fracture healing process. It has also been shown that in the early stages of bone repair, Sostdc1$^{-/-}$ mice (SOST is expressed only in the periosteum) lead to the recruitment of mesenchymal stem cells (MSCs) to the injured area, resulting in stronger bone formation than wild-type mice at day 21(Collette et al, 2016). It is shown that cells responding to the Wnt signaling pathway during fracture healing are not only OSX-positive cells, but there may be other cells responding to the Wnt signaling pathway or undergoing O-GlcNAcylation to promote fracture healing. The study also sheds light on how Wnt signaling regulates O-GlcNAcylation, although the precise details require further investigation. We screened metabolites through a metabolomic analysis and found that the substrate of HBP, UDP-GlcNAc, was enriched following the administration of rhWnt3a. In agreement with this finding, the first rate-limiting enzyme of HBP, GFAT1, was found to increase rapidly within 6 h of rhWnt3a treatment and increase continuously thereafter. According to previously published studies, GFAT1 is regulated at different levels, including through the modulation of mRNA and protein expression (Chaveroux et al, 2016; Manzari et al, 2007; Moloughney et al, 2016; Sayeski et al, 1997; Wang et al, 2014), post-translational modifications (Gelinas et al, 2018; Moloughney et al, 2018; Ruegenberg et al, 2021; Zhou et al, 1998; Zibrova et al, 2017), and allosteric regulation by metabolites (Mouilleron et al, 2008; Ruegenberg et al, 2020). Among these mechanisms of regulation, GFAT1 can also be transcriptionally regulated by Sp1 (Sayeski et al, 1997), as well as spliced by X-box-binding protein 1 (Xbp1s) (Wang et al, 2014), C-MYC (Liu et al, 2019), and ATF4 (Chaveroux et al, 2016) under different challenges such as glucose deprivation and hypoxia. Since Wnt3a rapidly induces GFAT1 overexpression, it is most likely through a post-translational modification. AMPK phosphorylates GFAT1 at Ser$^{243}$, which results in diminishing GFAT1 activity (Eguchi et al, 2009; Zibrova et al, 2017). In addition, the level of phosphorylation of GFAT1 at Ser$^{243}$ is also regulated by mTORC2 when proliferating cancer cells are submitted to either glucose or glutamine-limited conditions (Moloughney et al, 2018). However, in opposition to the AMPK effect, Ser$^{243}$ phosphorylation by mTORC2 stabilizes GFAT1 during nutrient-limited conditions, enabling the maintenance of flux through HBP (Moloughney et al, 2018). PKA-mediated phosphorylation at Ser$^{205}$ stabilizes the GFAT1 while impeding UDP-GlcNAc feedback inhibition, which leads to higher enzymatic activity (Zibrova et al, 2017). We examined the levels of protein expression of the PKA target phospho-CREB at Ser$^{133}$, phospho-AMPK at Thr$^{172}$, and the mTORC2 target phospho-AKT at Ser$^{473}$, in response to rhWnt3a. Upon Wnt stimulation, phospho-CREB, phospho-AMPK, and phospho-AKT all increased with different levels. The negative regulation of GFAT1 that phospho-AMPK induced likely reflects a fine-tuning regulation of HBP when GFAT1 increases. Compared with phospho-AKT, the change in phospho-CREB is more profound (Fig. 7B). And the pharmacological blocking experiment also demonstrated the essential role of PKA in GFAT1 regulation (Fig. 7D), but not OGT and OGA (Fig. EV4E). As reported, rhWnt3a significantly activates the mTOR pathway within 6 h (Fig. EV4J). Our data do not exclude the requirement of mTOR in regulating O-GlcNAc modification, since the mTOR inhibitor Torin1 also decreases GFAT1 protein level, but MK2206 has no effect (Fig. EV4K,L). The detailed mechanism behind this, however, merits further investigation in future studies.

Ca$^{2+}$ and PKA interact closely (Grewal et al, 2000; Ni et al, 2011); for instance, through calcium-mediated phosphorylation of CREB through the PKA-RAP1-ERK pathway. Studies have indicated that the β-catenin-independent Wnt3a pathway can also influence intracellular calcium flux. Wnt3a rapidly increases miniature synaptic currents through a mechanism involving Ca$^{2+}$ mobilization (Avila et al, 2010). Furthermore, Wnt3a inhibits

chondrogenesis in MSCs, via CaMKKII-mediated non-canonical Wnt signaling (Qu et al, 2013). In our study, we detected an increase of $Ca^{2+}$ flux in response to rhWnt3a, and blocking CaMKKII functionally blunted GFAT1 overexpression and O-GlcNAcylation. The inhibition of CaMKKII also decreased phospho-CREB levels, indicating a regulatory effect on the PKA pathway. Therefore, Wnt3a regulation of GFAT1 depends on $Ca^{2+}$ and PKA.

In addition to the acute stimulation by Wnt, we also showed that prolonged Wnt3a treatment results in an increase of O-GlcNAcylation through a β-catenin-dependent manner. As O-GlcNAcylation stabilizes β-catenin at $Thr^{41}$, it is reasonable to hypothesize that Wnt3a-induced O-GlcNAcylation also benefits the transduction of β-catenin-dependent Wnt signaling. Therefore, the increases in protein levels of glycolytic enzymes that we observed upon prolonged rhWnt3a treatment are likely due to the transcriptional regulation of β-catenin. Nonetheless, potential target proteins that undergo O-GlcNAcylation need further investigation. Our data indicated that PDK1, a regulator of glycolysis, possesses a $Ser^{174}$ O-GlcNAc site which, if mutated, can cause its degradation. However, this S174A mutation only decreases instead of fully diminishes the O-GlcNAc modification on PDK1 (Fig. 8E). This data highly recommends multiple O-GlcNAc sites regulate PDK1 function, which should be investigated later in more detail. Notably, the mutation of PDK1 also leads to lower levels of glycolysis and osteogenesis, which provides more evidence regarding the mechanism whereby Wnt regulates osteogenesis. In addition to PDK1, O-GlcNAcylation of other glycolysis enzymes, such as HK2, PFK1, PKM1/2, and LDHA were also detected (Fig. EV5C), which were consistent with reports in other tissue (Bacigalupa et al, 2018), suggesting that O-GlcNAcylation is a general regulatory mechanism of glycolysis.

Taken together, this study demonstrates that O-GlcNAc glycosylation orchestrates Wnt-stimulated bone formation and fracture healing, which provides a promising target for the development of future therapies against osteoporosis.

# Methods

## Mouse

Our study exclusively examined male mice. It is unknown whether the findings are relevant for female mice. All mouse procedures had the approval of the Ethics Committee of West China Hospital of Stomatology (batch number: WCHSIRB-D-2022-548). $Ogt^{flox/flox}$ and Osx-Cre mouse lines were as previously described (Rodda and McMahon, 2006; Shafi et al, 2000). To repress Cre activity, designated breeders were fed doxycycline-containing drinking water (500 μg/ml, 2% sucrose). Mice aged 21 days or 2 months (for fracture studies) were injected intraperitoneally with PBS or Scl-Ab (Anti-SOST antibody) (Chemstan, CSD00275, China) at 10 mg/kg every 4 days (Holdsworth et al, 2018) while drinking normal water, and harvested at the indicated times. For femoral fractures, an incision was made in the skin on the anterolateral aspect of the distal femur, exposing it to the mid-shaft. With the knee flexed, a sterile 23 G needle was inserted into the marrow cavity and a scalpel was used to produce the fracture at the midshaft of the femur.

## μCT analysis

The femurs were scanned with μCT 45 (Scanco Medical AG) and with the following key parameters: voxel size 13.1 $μm^3$, X-ray tube potential 55 kVp, X-ray intensity 145 μA, integration time 250 ms. To quantify trabecular bone parameters, 100 μCT slices (1.31 mm) immediately below the distal femoral growth plate were assessed. To quantify cortical bone parameters, 50 μCT slices (0.655 mm) of the mid-diaphyseal of the femur were analyzed. The thresholds for both analysis and 3D reconstruction were 300. For femoral fracture analysis and 3D reconstruction, contours were drawn around the entire callus margin. The threshold was set to 200.

## Bone histomorphology

For dynamic analysis, mice were injected intraperitoneally with Calcein green (Sigma, C0875) at a dose of 10 mg/kg at 7 days before euthanasia and with Alizarin complexone (Sigma, A3882) at a dose of 30 mg/kg at 2 days before euthanasia. H&E, Masson's trichrome, and TRAP stains were carried out on the paraffin sections according to the standard protocol.

## Immunofluorescence staining

Frozen sections of 6 μm thickness were performed on tibiae. After three PBS washes, slides were incubated with blocking solution (Antibody Diluent, Thermo Fisher Scientific) for 1 h and primary antibody overnight. The next day, PBS-washed slides were immersed in fluorescent secondary antibodies for 1 h. The antibodies involved in this part were listed below: anti-OSX antibody (Abcam, ab22552), anti-OGT antibody (Abcam, ab177941), anti-O-GlcNAc (RL2) antibody (Abcam, ab2739), Alexa Fluor 488-conjugated secondary antibody (Thermo Fisher Scientific, A11008), Alexa Fluor 647-conjugated secondary antibody (Abcam, ab150107). Finally, the slides were exposed to DAPI for 5 min and sealed with an anti-fluorescence quenching sealer. An Olympus SpinSR10 confocal microscope was used to acquire the images.

## Cell culture

The ST2 (RRID: CVCL_2205) cell line was grown in α-MEM (Gibco), the M2-10B4 (RRID: CVCL_5794) cell line was cultured in RPMI (Gibco), and the C3H/10T1/2 (RRID: CVCL_0190) and HEK-293T (RRID: CVCL_0063) cell lines were cultivated in DMEM (Gibco), which was complemented with 10% fetal bovine serum (FBS) (Zeta) and 1% Pen-Strep (P/S) (Gibco). Recombinant Human Wnt-3a Protein (rhWnt3a) (R&D Systems, 5036-WN-010/CF) was dosed to ST2 cells at 50 ng/ml. Unless otherwise specified, OSMI (Abcam, 20 μM), TMG (APExBIO, 10 μM), DON (APExBIO, 50 μM), NaCl (Sigma, 20 mM), LiCl (Sigma, 20 mM), XAV939 (APExBIO, 40 nM), 2-DG (APExBIO, 5 mM), H89 (APExBIO, 10 μM), STO-609 (MCE, 40 μM), Torin1 (APExBIO, 2 μM), MK2206 (APExBIO, 10 μM), IWP-L6 (APExBIO, 10 μM, 20 μM, 50 μM) and Compound C (APExBIO, 20 μM) were supplied to the treatment medium as indicated. The OGT-overexpressing adeno-virus with HA tag (pADV-mCMV-OGT-HA-T2A-mCherry, Ad-HA-OGT) was obtained from OBiO Technology Co., Ltd. (Shanghai, China), and ST2 cells were infected with adenovirus at

a confluence of about 70% for 24 h with a MOI of 40. For transient transfection, ST2 cells were transfected with a specific siRNA by means of Lipofectamine™ 3000 Transfection Reagent (Invitrogen) at ~70% confluence. All small interference RNAs (siRNAs) were synthesized by Hanbio Biotechnology Co., Ltd. (Shanghai, China). The siRNA sequences are shown in Appendix Table S3.

## Alkaline phosphatase (ALP) and Alizarin Red S (ARS) stains

ST2 cells were plated in 12-well plates, rinsed twice with chilled PBS, and fixed with 4% paraformaldehyde for 10 min. Subsequently, ALP and ARS stains were administered according to Beyotime (C3206, China) and Solarbio (1%, pH 4.2, G1452, China) kits, respectively. Cells were visualized with an Olympus system.

## RNA extraction and real-time quantitative PCR (RT-qPCR)

TRIzol reagent (Thermo Fisher Scientific) was used to extract total RNA. Extracted RNA was reverse transcribed into cDNA by reverse transcription kit (Vazyme, R223-01, China), followed by RT-qPCR using ChamQTM Universal SYBR® qPCR Master Mix kit (Vazyme, China). Primer sequences used are shown in Appendix Table S4. Using β-actin as a standard, the relative expression of target genes was evaluated by the $2^{-\Delta\Delta Ct}$ method.

## RNA sequencing

For RNA-Seq experiments, ST2 cells were treated with vehicle, OSMI (20 μM), or rhWnt3a (50 ng/ml) for 72 h before obtaining total RNA. Library construction and sequencing were accomplished by Novogene Co., Ltd. Differential expression analysis was run by the DESeq2 R package (1.20.0) with a threshold of $p$ value < 0.05 and |log2(foldchange)| > 0.5. Gene Ontology (GO) enrichment analysis was performed by clusterProfiler R package (v4.8.3), and GSEA analysis was completed by the local version of the GSEA analysis tool (https://www.gsea-msigdb.org/gsea/index.jsp). All heatmaps were done through the pheatmap package (v1.0.12) in R software (v4.3.0).

## Western blot

The primary antibodies for western blot were listed below: anti-O-GlcNAc (RL2) (Abcam, ab2739, 1:1000), anti-β-Actin (Beyotime, AF5003, 1:1000), anti-GFAT1 (Abcam, ab125069, 1:1000), anti-OGT (Abcam, ab177941, 1:1000), anti-OGA (Abcam, ab124807, 1:1000), anti-HK2 (Cell Signaling Technology, #2867, 1:1000), anti-PFKFB3 (Abcam, ab06699, 1:1000), anti-PFK1 (Novus Biologicals, JU53-31, 1:1000), anti-PKM1/2 (Cell Signaling Technology, #3190, 1:1000), anti-LDHA (Proteintech, 19987-1-AP, 1:2000), anti-PDK1 (Enzo, ADI-KAP-PK112-D, 1:1000), anti-β-catenin (ZEN BIO, R23616, 1:500), anti-pCREB (Ser133) (Cell Signaling Technology, #9198, 1:1000), anti-pAMPKα (Thr172) (Cell Signaling Technology, #2531, 1:1000), anti-AMPKα (Cell Signaling Technology, #2532, 1:1000), anti-pAKT (Ser473) (Cell Signaling Technology, #9271, 1:1000), anti-CaMKII (Proteintech, 11549-1-AP, 1:1000), anti-Flag (Sigma, F1804, 1:1000), anti-AKT (Cell Signaling Technology, #9272, 1:1000), anti-S6 (Cell Signaling Technology, #2217, 1:1000), anti-pS6 (Ser240/244) (Cell Signaling Technology, #5364, 1:1000),

anti-Rictor (ZEN BIO, 220460, 1:500), anti-pPKC (Thr638) (ZEN BIO, R22939, 1:500), anti-Raptor (Cell Signaling Technology, #2280, 1:1000), anti-pRaptor (Ser792) (Cell Signaling Technology, #2083, 1:1000), anti-Ubiquitin (Cell Signaling Technology, #43124, 1:1000). The next day, the PVDF membranes were incubated with Goat Anti-Rabbit IgG HandL (HRP) (Abcam, ab6721, 1:5000) or Goat Anti-Mouse IgG LCS (HRP) (Abbkine Scientific, A25012, 1:2000) secondary antibody for 1 h at room temperature. Blots were visualized using an ELC kit (Vazyme, E412-01, China) and captured by ChemiScope (Clinx, China). Protein was quantized by Image J software with β-Actin as the internal reference protein.

## Targeted metabonomics analysis

The Targeted metabonomics (important metabolites in the tricarboxylic acid cycle, glycolytic pathway, and oxidative phosphorylation processes) analysis was done by Shanghai Applied Protein Technology Co., Ltd. (Shanghai, China). Quantitative analysis was performed using MultiQuant or Analyst software.

## Glucose uptake and lactate production assay

After ST2 cells were treated for a specific time, a cell culture medium was collected for glucose and lactate assays. Concentration measurements were performed using glucose (Sigma, GAHK20) and lactate (Eton Biosciences, 1200011002) kits. Specific methods were described previously (You et al, 2022). Protein concentration was used for normalization.

## Seahorse assay

Oxygen consumption rate (OCR) and extracellular acidification rate (ECAR) measurements were performed using the Seahorse XFe24 Analyzer (Agilent) according to the manufacturer's operator's manual. ST2 cells for the indicated treatments were seeded on 24-well Seahorse assay plates at a density of 40,000 cells/well. After 2 h, the cell medium was changed to Seahorse XF assay solution containing 5.5 mM glucose, 2 mM glutamine, 0.1 mM pyruvate, and 5 mM HEPES. For the Mito Stress experiment, the compounds added to the probe plate were Oligomycin (1.5 μM), Carbonyl cyanide-4 (trifluoromethoxy) phenylhydrazone (FCCP) (1.5 μM), Rotenone (1 μM) and Antimycin (1 μM). Seahorse data analysis was performed using Agilent Seahorse Wave software.

## Intracellular Ca$^{2+}$ signal detection

ST2 cells were incubated with 4 μM Fluo-4/AM (Shanghai Maokang Biotechnology Co, Ltd, MX4504) for 30 min at 37 ℃ after overnight serum starvation, then washed three times with Hanks' buffer containing 20 mM Hepes (HHBS, w/o Ca$^{2+}$, Mg$^{2+}$) and incubated for 30 min, followed by fluorescence microscopy. rhWnt3a (50 ng/ml), ionomycin (1 μM), and corresponding vehicle were injected 2 min after the focus image. Time-lapse images were acquired every 3 s. Statistical analysis of intracellular Ca$^{2+}$ signal fluorescence intensity using ImageJ.

## Co-immunoprecipitation

Protein A + G Agarose (Beyotime, P2012) was used following the producer's instructions. For antibody conjugation, 1 μg of primary antibody was used for immunoprecipitation with slow shaking at

4 °C overnight. The conjugate primary antibodies used were as shown below: anti-O-GlcNAc (RL2) (Thermo Fisher Scientific, MA1-072), and anti-Flag (Thermo Fisher Scientific, F1804). Mouse IgG (Beyotime, A7028) was used as a control IgG.

## Prediction of PDK1 O-GlcNAcylation sites

PDK1 O-GlcNAcylation sites were predicted on the YinOYang - 1.2 website (https://services.healthtech.dtu.dk/services/YinOYang-1.2/).

## Generation of stable cell lines

HEK-293T cells were co-transfected with a lentiviral vector containing psPAX2 and pMD2.G using lipo8000 (Beyotime, C0533-0.5 ml). Viral supernatants were harvested at 48 and 72 h, filtered through a 0.45-μm filter (Millipore; SLHP033RS), and 8 μg/ml polybrene was added to infect ST2 cells. To obtain stable transient cell lines, cells were infected with lentivirus for 48 h and then screened with puromycin dihydrochloride (APExBIO, B7587) for 2 weeks. pLV[Exp]-Puro-EF1A>3xFLAG/mPdk1[NM_172665.5](ns)*:T2A:EGFP acquired from Vector Builder (VB220828-1259bcu). Site-specific mutagenesis of S174A was generated using the QuikChange® II XL mutagenesis kit (Agilent, #200522) following the instructions of the manufacturer.

## Determination of PDK1 half-life

Stable cell lines of PDK1 WT or S174A mutation were exposed to CHX for the indicated times (0, 2, 6, and 24 h) prior to protein harvest, and PDK1 protein levels were measured with anti-Flag antibody to calculate relative half-life.

## Statistical analysis

GraphPad Prism 8.0 was used for the analysis of the data. Data were given as mean ± standard deviation (SD). Quantitative data were calculated using two-tailed Student's t-test, one-way ANOVA, or two-way ANOVA. The statistical significance was defined as $p < 0.05$.

# Data availability

RNA-seq data have been deposited at the Gene Expression Omnibus under accession code GSE251951.

The source data of this paper are collected in the following database record: biostudies:S-SCDT-10_1038-S44319-024-00237-z.

# Peer review information

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

## Acknowledgements

We thank Dr. Bei Yin and Dr. Feng Lou for the assistance with the phenotypic analysis of the mice. This work was supported by the National Natural Science Foundation of China (Grant No: 82071091 to YS, 82001040 to BL), the National Science Fund for Distinguished Young Scholars of China (Grant No: 81825005 to LY), West China Hospital of Stomatology Sichuan University (Grant No: RD-03-202304 and QDJF2021-1 to YS).

## Author contributions

**Chengjia You**: Data curation; Software; Validation; Investigation; Visualization; Methodology; Writing—original draft. **Fangyuan Shen**: Investigation; Visualization; Methodology. **Puying Yang**: Investigation; Visualization; Methodology. **Jingyao Cui**: Investigation; Methodology. **Qiaoyue Ren**: Methodology. **Moyu Liu**: Methodology. **Yujie Hu**: Methodology. **Boer Li**: Methodology. **Ling Ye**: Conceptualization; Supervision; Writing—review and editing. **Yu Shi**: Conceptualization; Supervision; Writing—review and editing.

Source data underlying figure panels in this paper may have individual authorship assigned. Where available, figure panel/source data authorship is listed in the following database record: biostudies:S-SCDT-10_1038-S44319-024-00237-z.

## Disclosure and competing interests statement

The authors declare no competing interests.

# Expanded View Figures

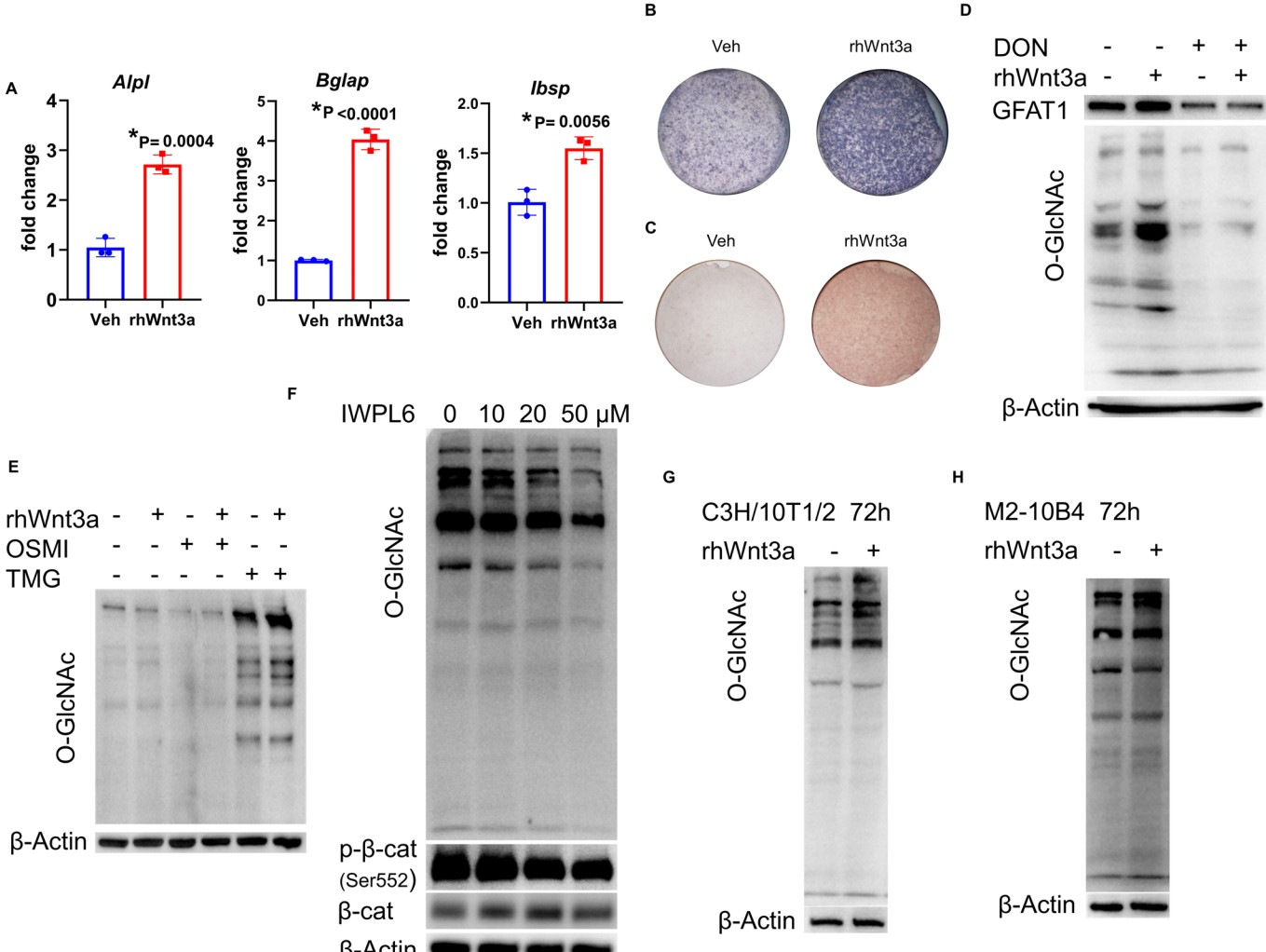

**Figure EV1. Wnt3a increases protein O-GlcNAcylation that mediates Wnt-induces osteogenesis in vitro.**

(related to Figs. 1 and 2) (**A**) ST2 cells were treated with 50 ng/ml rhWnt3a for 3 days and relative expression of osteogenic genes was tested by qPCR. Each dot represented one single experiment. Error bars: mean ± SD. *$p < 0.05$; $n = 3$, biological replicates (two-tailed Student's t-test). (**B**) ALP staining, and (**C**) Alizarin red S staining were performed after rhWnt3a treatment for 3 days and 14 days, respectively. (**D**) ST2 were pretreated with GFAT antagonist DON for 6 h and then administrated with rhWnt3a for 2 days. GFAT1 and O-GlcNAc levels were detected by Western blotting. (**E**) ST2 were pretreated with OGT and OGA inhibitors, OSMI and TMG, respectively for 6 h and then administrated with rhWnt3a for 2 days. O-GlcNAc levels are detected by Western blotting. (**F**) ST2 cells were pretreated with IWPL6 for 2 days, and O-GlcNAc, phospho-β-catenin(Ser552) and total β-catenin levels were detected by Western blotting. (**G, H**) O-GlcNAcylation expression levels of total proteins in C3H/10T1/2 and M2-10B4 cells after 72 h induction by rhWnt3a.

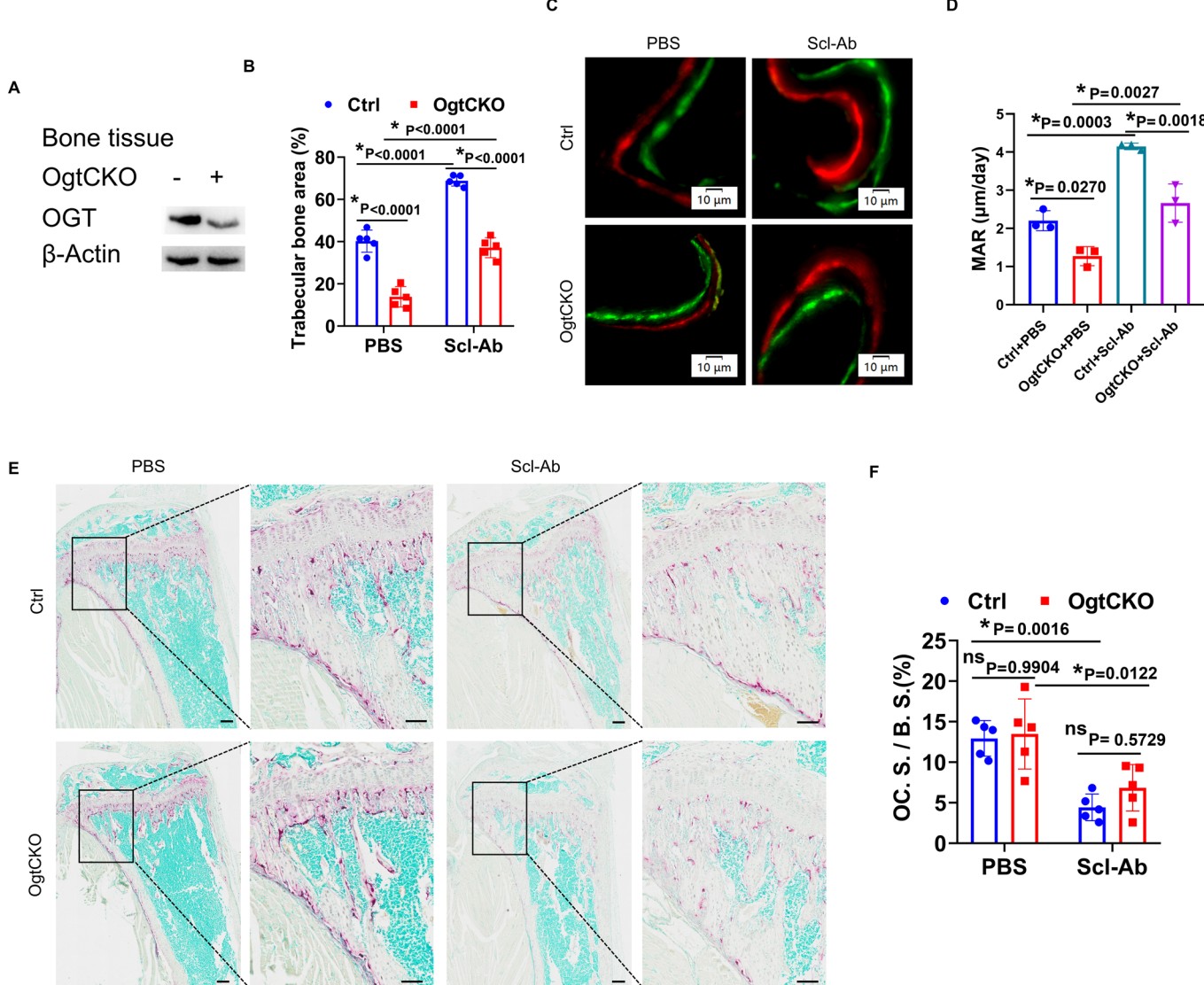

**Figure EV2. Deletion of O-GlcNAcylation in osteoblast-lineage cells diminishes Wnt-induced bone formation.**

(related to Fig. 3) (**A**) An examination of OGT protein was conducted in femoral and tibial diaphysis tissues from Ctrl mice and OgtCKO mice. (**B**) Quantification of proximal tibial trabeculae in Fig. 3H. (**C**) Representative images of calcein–alizarin red double labeling in the trabecular bone of femurs. Scale bar, 10 μm. (**D**) The quantification of the Mineral Apposition Rate (MAR) of trabecular bone is shown. (**E**) TRAP staining for the tibia from Ctrl or OgtCKO mice with or without Scl-Ab injection, scale bar, 200 μm. Boxed area is shown at a high magnification to the right, scale bar, 100 μm. (**F**) Osteoclast (TRAP-positive multinucleated cells) surface normalized to bone surface (OC. S./B. S.) is shown. The ROI was around 0.09 mm² region (300 × 300 μm) underneath the growth plate. Each dot represented one animal. Error bars: mean ± SD. *$p < 0.05$, ANOVA followed by Tukey's multiple comparisons test.

A

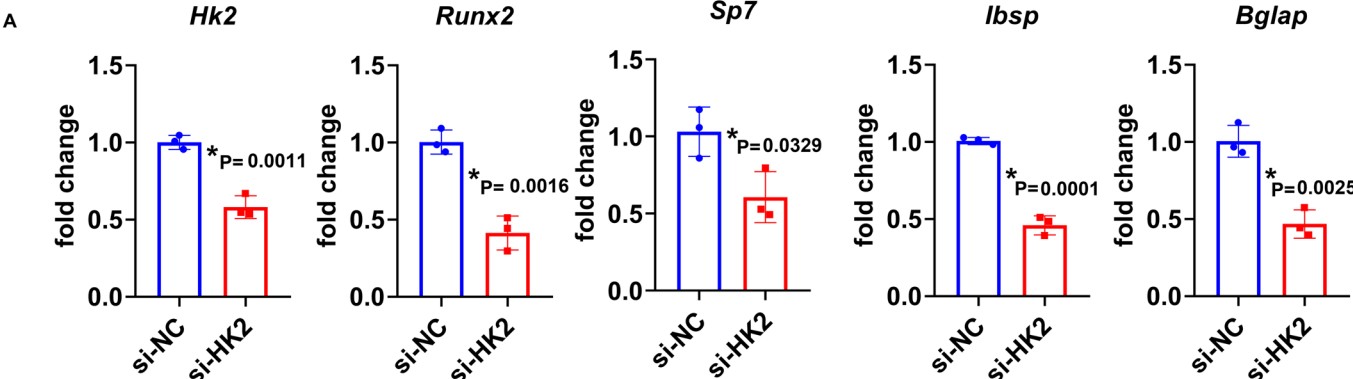

B

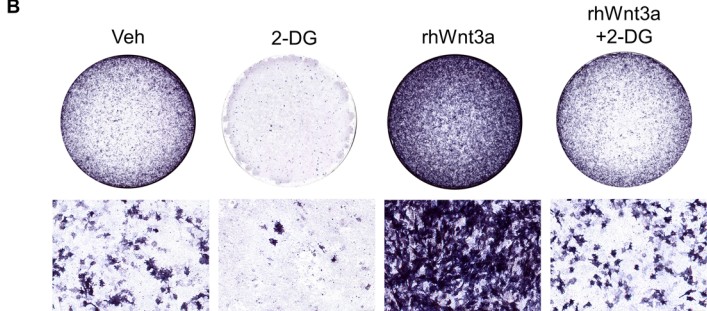

**Figure EV3. O-GlcNAcylation is indispensable for Wnt3a-increased aerobic glycolysis.**

(related to Figs. 5 and 8) (A) Osteoblastic gene detection by qPCR in the absence of Hk2. Each dot represented one single experiment. Error bars: mean ± SD. *$p < 0.05$; $n = 3$ (two-tailed Student's t-test). (B) ST2 cells were pretreated with 2-DG for 1 day and then exposed to rhWnt3a for 3 additional days. Then ALP staining was performed.

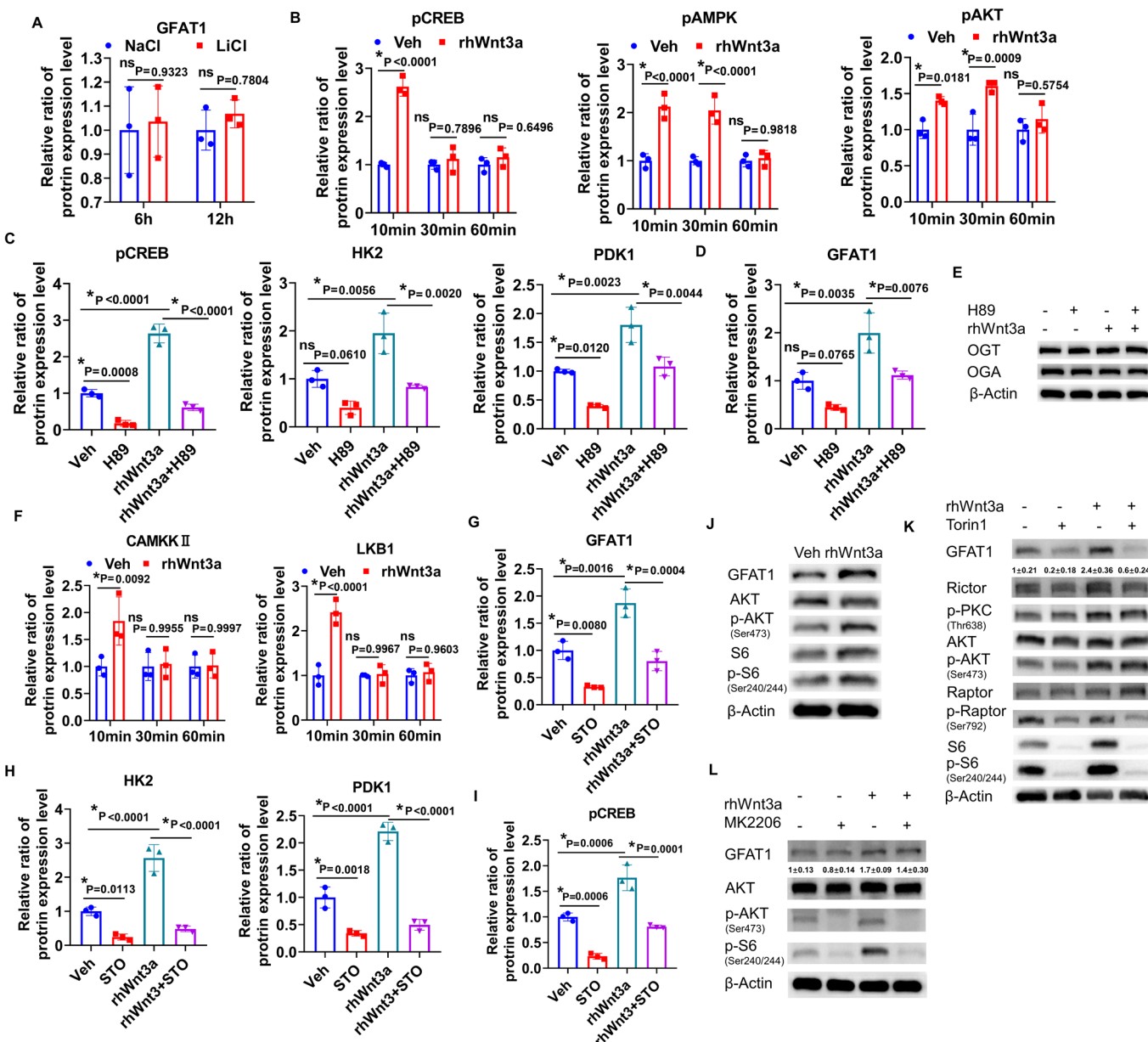

**Figure EV4.  Wnt rapidly increases O-GlcNAcylation via the Ca²⁺-PKA-GFAT1 axis.**

(related to Fig. 7) (**A**) Quantification of GFAT1 protein shown in Fig. 7A. (**B**) Quantification of pCREB, pAMPK, and pAKT proteins shown in Fig. 7B. (**C**) Quantification of pCREB, HK2, and PDK1 proteins shown in Fig. 7C. (**D**) Quantification of GFAT1 protein shown in Fig. 7D. (**E**) ST2 cells were treated with rhWnt3a, H89, or both for 6 h. OGT and OGA expression levels were detected. (**F**) Quantification of CAMKII and LKB1 proteins shown in Fig. 7F. (**G**) Quantification of GFAT1 protein shown in Fig. 7G. (**H**) Quantification of HK2 and PDK1 proteins shown in Fig. 7H. (**I**) Quantification of pCREB protein shown in Fig. 7K. (**J**) The mTOR downstream targets expression in response to rhWnt3a. (**K**) The mTOR inhibition decreases GFAT1 protein level. (**L**) The GFAT1 expression detection after the mTORC2 inhibition. All data are shown as mean ± SD, $n = 3$, biological replicates. *$p < 0.05$, (**A, B, F**) two-way ANOVA followed by Sidak's multiple comparisons test, (**C, D, G, H and I**) one-way ANOVA followed by Tukey's multiple comparisons test.

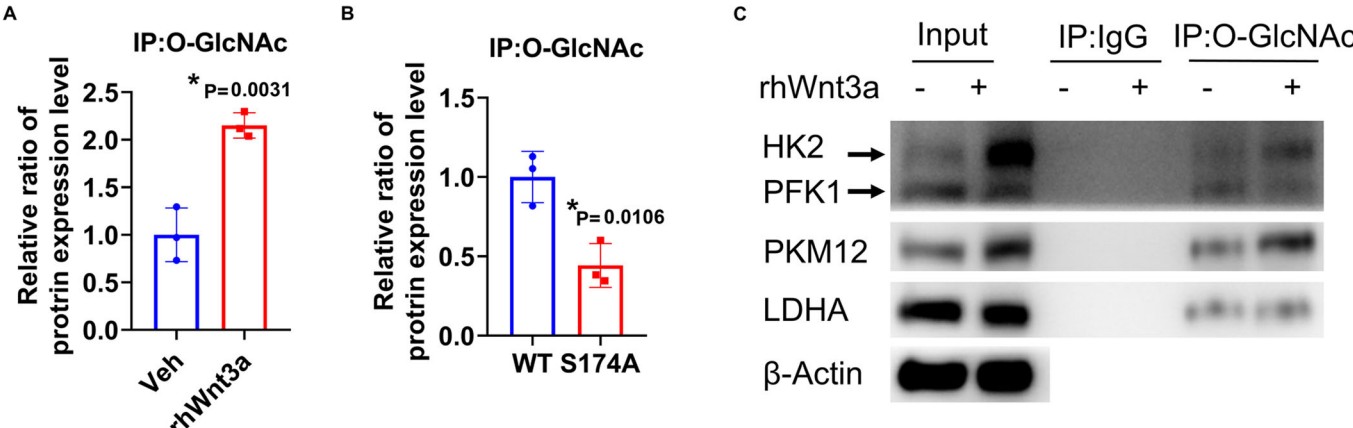

**Figure EV5.  O-GlcNAcylation on the Ser[174] site stabilizes PDK1 and facilitates Wnt3a-induced osteogenesis.**

(related to Fig. 8) (**A**) Quantification of the IPed O-GlcNAc protein in Fig. 8B. (**B**) Quantification of the IPed O-GlcNAc protein in Fig. 8E. (**C**) Immunoprecipitation with endogenous O-GlcNAc and detection with HK2, PFK1, PKM12, and LDHA, respectively. Error bars: mean ± SD. *$p < 0.05$; $n = 3$, biological replicates (two-tailed Student's t-test).

