## [Peer Review File · EMBO Reports]

O-GlcNAcylation mediates Wnt-stimulated bone formation by rewiring aerobic glycolysis

Chengjia You, Fangyuan Shen, Puying Yang, Jingyao Cui, Qiaoyue Ren, Moyu Liu, Yujie Hu, Boer Li, Ling Ye, and Yu Shi

Corresponding author(s): Yu Shi (yushi1105@scu.edu.cn) , Ling Ye (yeling@scu.edu.cn)

Review Timeline:

Submission Date:	22nd Mar 24
Editorial Decision:	15th Apr 24
Revision Received:	16th Jun 24
Editorial Decision:	17th Jul 24
Revision Received:	31st Jul 24
Accepted:	13th Aug 24

Editor: Achim Breiling

Transaction Report:

Dear Prof. Shi,

Thank you for the submission of your manuscript to EMBO reports. I have now received the reports from the three referees that were asked to evaluate your study, which can be found at the end of this email.

As you will see, all referees have several comments, concerns, and suggestions, indicating that a major revision of the manuscript is necessary to allow publication of the study in EMBO reports. As the reports are below, and all the concerns need to be addressed, I will not detail them here.

Given the constructive referee comments, I would like to invite you to revise your manuscript with the understanding that the concerns of the referees must be addressed in the revised manuscript or in a detailed point-by-point response. Acceptance of your manuscript will depend on a positive outcome of a second round of review. It is EMBO reports policy to allow a single round of revision only and acceptance of the manuscript will therefore depend on the completeness of your responses included in the next, final version of the manuscript.

- 1) a .docx formatted version of the final manuscript text (including legends for main figures, EV figures and tables), but without the figures included. Figure legends should be compiled at the end of the manuscript text.
- 2) individual production quality figure files as .eps, .tif, .jpg (one file per figure), of main figures and EV figures. Please upload these as separate, individual files upon re-submission.

The Expanded View format, which will be displayed in the main HTML of the paper in a collapsible format, has replaced the Supplementary information. You can submit up to 5 images as Expanded View. Please follow the nomenclature Figure EV1, Figure EV2 etc. The figure legend for these should be included in the main manuscript document file in a section called Expanded View Figure Legends after the main Figure Legends section. Additional Supplementary material should be supplied as a single pdf file labelled Appendix. The Appendix should have page numbers and needs to include a table of content on the first page (with page numbers) and legends for all content. Please follow the nomenclature Appendix Figure Sx, Appendix Table Sx etc. throughout the text, and also label the figures and tables according to this nomenclature.

- 4) a complete author checklist, which you can download from our author guidelines (<https://www.embopress.org/page/journal/14693178/authorguide>). Please insert page numbers in the checklist to indicate where the requested information can be found in the manuscript. The completed author checklist will also be part of the RPF.

- 5) that primary datasets produced in this study (e.g. RNA-seq, ChIP-seq, structural and array data) are deposited in an

appropriate public database. If no primary datasets have been deposited, please also state this in a dedicated section (e.g. 'No primary datasets have been generated and deposited'), see below.

The accession numbers and database should be listed in a formal "Data Availability" section (placed after Materials & Methods) that follows the model below. This is now mandatory (like the COI statement). Please note that the Data Availability Section is restricted to new primary data that are part of this study. This section is mandatory. As indicated above, if no primary datasets have been deposited, please state this in this section

Data availability

7) Our journal encourages inclusion of *data citations in the reference list* to directly cite datasets that were re-used and obtained from public databases. Data citations in the article text are distinct from normal bibliographical citations and should directly link to the database records from which the data can be accessed. In the main text, data citations are formatted as follows: "Data ref: Smith et al, 2001" or "Data ref: NCBI Sequence Read Archive PRJNA342805, 2017". In the Reference list, data citations must be labelled with "[DATASET]". A data reference must provide the database name, accession number/identifiers and a resolvable link to the landing page from which the data can be accessed at the end of the reference. Further instructions are available at: <http://www.embopress.org/page/journal/14693178/authorguide#referencesformat>

8) Regarding data quantification and statistics, please make sure that the number "n" for how many independent experiments were performed, their nature (biological versus technical replicates), the bars and error bars (e.g. SEM, SD) and the test used to calculate p-values is indicated in the respective figure legends (also for EV figures and all those in an Appendix). Please also check that all the p-values are explained in the legend, and that these fit to those shown in the figure. Please provide statistical testing where applicable. Please avoid the phrase 'independent experiment', but clearly state if these were biological or technical replicates. Please also indicate (e.g. with n.s.) if testing was performed, but the differences are not significant. In case n=2, please show the data as separate datapoints without error bars and statistics. See also: <http://www.embopress.org/page/journal/14693178/authorguide#statisticalanalysis>

9) Please add scale bars of similar style and thickness to microscopic images, using clearly visible black or white bars (depending on the background). Please place these in the lower right corner of the images themselves. Please do not write on or near the bars in the image but define the size in the respective figure legend.

10) Please also note our reference format:

12) We now use CRedit to specify the contributions of each author in the journal submission system. CRedit replaces the author contribution section. Please use the free text box to provide more detailed descriptions and do not provide your final manuscript text file with an author contributions section. See also our guide to authors: <https://www.embopress.org/page/journal/14693178/authorguide#authorshippinguidelines>

13) We would encourage you to use 'Structured Methods', our new Materials and Methods format. According to this format, the

Materials and Methods section should include a Reagents and Tools Table (listing key reagents, experimental models, software, and relevant equipment and including their sources and relevant identifiers), uploaded as separate file, followed by a Methods and Protocols section in which we encourage the authors to describe their methods using a step-by-step protocol format with bullet points, to facilitate the adoption of the methodologies across labs. More information on how to adhere to this format as well as downloadable templates (.doc or .xls) for the Reagents and Tools Table can be found in our author guidelines (section 'Structured Methods'):

14) Please add up to five keywords to the manuscript and provide the abstract written in present tense. Please also order the manuscript sections like this, using these names:

Title page - Abstract - Keywords - Introduction - Results - Discussion - Methods - Data availability section - Acknowledgements - Disclosure and Competing Interests Statement - References - Figure legends - Expanded View Figure legends

I look forward to seeing a revised version of your manuscript when it is ready. Please let me know if you have questions or comments regarding the revision.

Please use this link to submit your revision: <https://embor.msubmit.net/cgi-bin/main.plex>

Yours sincerely,

Referee #1:

The authors have investigated the mechanisms for Wnt-induced bone formation. Through metabolomic and biochemical studies in osteoblast differentiation in vitro, they discovered the increase in HBP pathway in response to Wnt3a and further that the pathway activation is required for in vitro differentiation. They then performed genetic deletion studies to show that Ogt, key for o-linked glycosylation, is required for bone formation both under normal conditions and in response to anti-Sost. Subsequent biochemical and metabolic studies identified a strong link between Wnt and O-linked glycosylation and glycolysis in cell cultures via both b-catenin dependent (slow effect) and independent (fast effect) mechanisms. They finally described Pdk1 as a target of O-linked glycosylation that plays a role in the activation of both glycolysis and osteoblastogenesis in response to Wnt3a in vitro. Overall this is a substantial amount of work that is generally of good quality and supports the novel mechanism that Wnt signaling activates osteoblasts in part through increased glycosylation of key metabolic proteins to increase glycolysis. There are several issues that should be addressed.

Fig.3: The genetic deletion of Ogt in osteoblast lineage showed that the cortical bone effect by anti-sost was completely abolished whereas the effect on trabecular bone was maintained. The authors provided double labeling data for the cortical bone with shows virtual elimination of the anabolic effect of anti-sost, providing strong evidence for the conclusion that Wnt anabolic effect is critically dependent on Ogt. As anti-sost is known to both increase bone formation and suppress bone resorption, it is possible that the increased trabecular bone mass by anti-sost in the Ogt CKO mice reflected the anti-suppression effect that is likely still intact. They provided OC quantification in supplemental Fig. S4 but used few mice and the variations among mice were big. Do they have double labeling data for the trabecular bone? Serum markers for bone formation and resorption can also be informative.

Fig. 7B: Western blot quantification does not seem to match the signal intensity in figures. This should be double checked for accuracy.

Fig. 9: The model, if so desired, should be revised to clarify several things. 1) glycolysis is stimulated by Wnt signaling at multiple steps besides the Ldha step; 2) Pdk1 is only one of the possibly many targets of O-linked glycosylation that is stimulated by Wnt signalig; 3) Pdk1 does not directly stimulate pyruvate to lactate conversion, but rather works in the mitochondria to inhibit Pdh activity that converts pyruvate to acetyl-coA.

Referee #2:

In the manuscript by You et al. they investigate the role O-GlcNAcylation plays in bone formation. The authors demonstrate that treatment of ST2 cells with Wnt3a increases O-GlcNAcylation both in vitro and in vivo. Deletion of O-GlcNAcylation reduced Wnt stimulated osteoblast differentiation and fracture healing. O-GlycNAcylation diminished aerobic glycolysis and interfered with

bone formation. While this is a very complete study there are some controls that should be included to help the reader understand the rigor of the study.

1. There should be verification of reduced OGT expression in cells in OGTcKO mice.
2. The text should be modified to indicate that it is not osteoclast activity but osteoclast differentiation that was measured in supplementary figure 4.
3. In figure 4C and D the change in both BMD and % BV/TV between WT and KO +/- Scl antibody appears to be the same fold change (It appears when you look at the numbers in Table 3, supplemental). Actual numbers should be included in the text and not in supplemental data. If fold change between WT and OgtcKO +/- Scl antibody are similar than please modify the text in the results and add potential reasons for this result should be in the discussion.
4. OGA and OGT knockdown by siRNA do not seem very significant in Figure 5i. Knockdowns for all siRNA experiments should be quantitated.

Referee #3:

The authors show that the osteoanabolic influence of Wnt3a on mesenchymal ST2 cells is mediated, at least in part, by Wnt3a-dependent induction of O-GlcNacylation. They also identify one specifically modified protein (PDK1), where the Wnt3a-induced O-GlcNacylation at Ser174 is critical for osteoblastogenesis. By generating a mouse model with inducible osteoblast-specific inactivation of O-GlcNac transferase they further demonstrate the in vivo relevance of their findings, as these mice had in impaired osteoanabolic response towards Wnt activation by a sclerostin antibody. Overall, the results are convincingly presented, and they provide novel insights into a clinically relevant pathway controlling bone formation. There are however few issues that remain to be addressed.

Specific comments:

- 1) The first sentence of the abstract is somehow misleading, as the newly approved anabolic therapy for osteoporosis, i.e. Romosozumab, is blocking a Wnt signaling inhibitor (sclerostin), which should be clearly stated. Moreover, the potential influence of Wnt signaling on aerobic glycolysis is not relevant for this novel treatment approach.
- 2) The authors introduce many abbreviations, which is not very reader-friendly. At least they should provide a list of the most relevant abbreviations used throughout the manuscript.
- 3) The presentation of the surely important metabolite screening shown in Figure 1A should be modified, since the labeling is far too small to ensure readability of a printed version.
- 4) The experiment shown in Fig. S2 needs to be better explained in the Results section (the respective sentence is also incomplete).
- 5) There is no specific information regarding the antibody that was used to inhibit sclerostin. Did you authors use a murine sclerostin antibody?
- 6) Regarding the mouse data presented in Figure 3/4 the authors have to be precise about the controls, as there are several possibilities, also in terms of doxycyclin removal. Most importantly, whereas it is first stated in the Results section that Osx-Cre mice were used as controls, the legend of Figure 3 refers to WT mice as controls. Moreover, while it is stated in the text that "the cancellous bone mass was still increased by Slc-AB in the Ogt CKO mice", this particular statistical comparison is not displayed in Figure 3C-E. For the sake of clarity, it might be an option to provide the respective p-values in the text. Finally, it is not informative to show the HE and Masson staining in Figure 3G/H and Figure 4E/F without quantification.
- 7) Whereas the molecular results are generally convincing and confirmed by quantitative analysis, that data regarding the 6-hour response towards Wnt3a are less convincingly presented. Especially since the "dramatically increased intracellular calcium flux after 4 mins of rhWnt3a administration" shown in Figure 7E (red line) is not very evident, a quantification of the respective data should be provided. Moreover, there is no information about statistically significant differences regarding the Western blot results, and it might be more informative to provide this information by showing bar graphs. The same applies for the data shown in Figure 8B/E.
- 8) Although it is surely important to utilize Wnt3a and ST2 cells in order to understand the molecular influence of the Wnt signaling pathway for osteogenesis, the authors should at least critically discuss that there is so far no human genetic evidence for a specific role of WNT3a in human bone mass acquisition (unlike it is the case for WNT1). Another important issue that should be discussed and/or experimentally addressed relates to the cell type specificity of the observed influences. In other words, does every cell type react to Wnt3a by induced O-GlcNacylation, or is this specific for osteoblast progenitor cells?

Dear Editors,

I would like to express our sincere appreciation of the reviewers' constructive comments concerning our article entitled "O-GlcNAcylation Mediates Wnt-stimulated Bone Formation via Rewiring Aerobic Glycolysis in Osteoblast-lineage Cells". These comments are all valuable and helpful for improving our article. According to the reviewers' comments, we have made modifications to our manuscript. In this revised version, changes to our manuscript were all highlighted within the manuscript by using red-colored text. Point-by-point responses to the reviewers are listed below this letter.

Referee #1:

The authors have investigated the mechanisms for Wnt-induced bone formation. Through metabolomic and biochemical studies in osteoblast differentiation in vitro, they discovered the increase in HBP pathway in response to Wnt3a and further that the pathway activation is required for in vitro differentiation. They then performed genetic deletion studies to show that Ogt, key for o-linked glycosylation, is required for bone formation both under normal conditions and in response to anti-Sost. Subsequent biochemical and metabolic studies identified a strong link between Wnt and O-linked glycosylation and glycolysis in cell cultures via both b-catenin dependent (slow effect) and independent (fast effect) mechanisms. They finally described Pdk1 as a target of O-linked glycosylation that plays a role in the activation of both glycolysis and osteoblastogenesis in response to Wnt3a in vitro. Overall this is a substantial amount of

work that is generally of good quality and supports the novel mechanism that Wnt signaling activates osteoblasts in part through increased glycosylation of key metabolic proteins to increase glycolysis. There are several issues that should be addressed.

Fig.3: The genetic deletion of Ogt in osteoblast lineage showed that the cortical bone effect by anti-sost was completely abolished whereas the effect on trabecular bone was maintained. The authors provided double labeling data for the cortical bone with shows virtual elimination of the anabolic effect of anti-sost, providing strong evidence for the conclusion that Wnt anabolic effect is critically dependent on Ogt. As anti-sost is known to both increase bone formation and suppress bone resorption, it is possible that the increased trabecular bone mass by anti-sost in the Ogt CKO mice reflected the anti-suppression effect that is likely still intact. They provided OC quantification in supplemental Fig. S4 but used few mice and the variations among mice were big. Do they have double labeling data for the trabecular bone? Serum markers for bone formation and resorption can also be informative.

Response: Thanks for raising this important concern. We increased the number of mice to n = 5 and performed statistical analyses of TRAP staining, which showed that in the trabecular region, Scl-Ab reduced osteoclast area in both Ctrl and OgtCKO mice, however, there was no statistical difference in osteoclast area between Ctrl and OgtCKO mice with or without Scl-Ab administration (Fig. EV2 E and F). This data confirmed that the Scl-Ab suppressive effect on osteoclasts in OgtCKO mice is still intact.

Importantly, the data also indicated the difference in trabecular bone mass was not due to the alteration of bone resorption between Ctrl and OgtCKO.

Meanwhile, double-labeling data for trabecular bone are shown in Figure EV2 C and D. Consistent with the finding in cortical bone, the results showed that trabecular bone MAR was increased in Ctrl mice after Scl-Ab treatment, however, this increase was suppressed in OgtCKO mice which suggested Wnt anabolic effect was critically dependent on Ogt in both trabeculae and cortical bone.

Fig. 7B: Western blot quantification does not seem to match the signal intensity in figures. This should be double checked for accuracy.

Response: Thanks for your suggestion. We are sorry to cause this confusion. In this setting, we defined each Veh in the respective time point as the standard 1. The rhWnt3a treated group normalized to each Veh and received the statistics of fold changes. In this revision, we also clarified how we did quantification in the legends of the main text (line 971-972).

Fig. 9: The model, if so desired, should be revised to clarify several things. 1) glycolysis is stimulated by Wnt signaling at multiple steps besides the Ldha step; 2) Pdk1 is only one of the possibly many targets of O-linked glycosylation that is stimulated by Wnt signal; 3) Pdk1 does not directly stimulate pyruvate to lactate conversion, but rather works in the mitochondria to inhibit Pdh activity that converts pyruvate to acetyl-coA.

Response: Thank you very much for the reminder. We appreciate and agree with your comments. In the previous version, we tried to avoid more detailed info such as the role of Pdh in this process. In revision, besides discussing the multiple O-GlcNAc target proteins in the discussion (line 489-492), we have modified the graphical abstract. The revised version is as follows:

Referee #2:

In the manuscript by You et al. they investigate the role O-GlcNAcylation plays in bone formation. The authors demonstrate that treatment of ST2 cells with Wnt3a increases O-GlcNAcylation both in vitro and in vivo. Deletion of O-GlcNAcylation reduced Wnt stimulated osteoblast differentiation and fracture healing. O-GlycNAcylation diminished aerobic glycolysis and interfered with bone formation. While this is a very complete study there are some controls that should be included to help the reader understand the rigor of the study.

1. There should be verification of reduced OGT expression in cells in OGTcKO mice.

Response: Thank you for your suggestion. We performed the Western blotting experiment to compare the protein level of Ogt in the bone extract between Ctrl and OgtCKO mice. The validation of the knockdown efficiency of OGT is shown in Figure EV2 A.

2. The text should be modified to indicate that it is not osteoclast activity but osteoclast

differentiation that was measured in supplementary figure 4.

Response: Thank you for your suggestion. We have reorganized supplementary figure 4 into Figure EV2, and the “osteoclast activity” in the legend has been removed.

3. In figure 4C and D the change in both BMD and % BV/TV between WT and KO +/- Scl antibody appears to be the same fold change (It appears when you look at the numbers in Table 3, supplemental). Actual numbers should be included in the text and not in supplemental data. If fold change between WT and OgtcKO +/- Scl antibody are similar than please modify the text in the results and add potential reasons for this result should be in the discussion.

Response: Thank you for your question. In Figure 4, We labeled the p-value for each comparison on top of the bar graph to give a better idea of which was significant. We also put the actual numbers next to the bar graph in the main figure. Based on your kind suggestion, we also noticed a similar fold change between Ctrl and OgtCKO with or without Scl-Ab. Therefore, we modified our conclusion as: “The above data suggest that Wnt-mediated fracture healing is mediated by OGT, but not exclusively dependent upon OGT (line 235-236).” The corresponding discussion is added in lines 421 to 432.

4. OGA and OGT knockdown by siRNA do not seem very significant in Figure 5i.

Knockdowns for all siRNA experiments should be quantitated.

Response: In Figures 5I and 6K, we have added the relative quantification of siRNA knockdown efficiencies. In Figure 5I, we designated the quantification of rhWnt3a-siOGT-siOGA- (the first lane) as standard 1. The siOGT and siOGA were normalized to the first lane and received fold changes, respectively. In Figure 6K, we designated the quantification of rhWnt3a-; si-beta-Cat-1-; si-beta-Cat-3- (the first lane) as standard 1. The si-beta-Cat-1, -3 were normalized to the first lane and received fold changes, respectively (lane 2 and 3).

Referee #3:

The authors show that the osteoanabolic influence of Wnt3a on mesenchymal ST2 cells is mediated, at least in part, by Wnt3a-dependent induction of O-GlcNacylation. They also identify one specifically modified protein (PDK1), where the Wnt3a-induced O-GlcNacylation at Ser174 is critical for osteoblastogenesis. By generating a mouse

model with inducible osteoblast-specific inactivation of O-GlcNac transferase they further demonstrate the in vivo relevance of their findings, as these mice had an impaired osteoanabolic response towards Wnt activation by a sclerostin antibody. Overall, the results are convincingly presented, and they provide novel insights into a clinically relevant pathway controlling bone formation. There are however few issues that remain to be addressed.

Specific comments:

- 1) The first sentence of the abstract is somehow misleading, as the newly approved anabolic therapy for osteoporosis, i.e. Romosozumab, is blocking a Wnt signaling inhibitor (sclerostin), which should be clearly stated. Moreover, the potential influence of Wnt signaling on aerobic glycolysis is not relevant for this novel treatment approach.

Response: Thank you for the kind reminder. We have made a revision to the first paragraph of the abstract (line 14 to 17).

- 2) The authors introduce many abbreviations, which is not very reader-friendly. At least they should provide a list of the most relevant abbreviations used throughout the manuscript.

Response: Thank you for your suggestion. We have added it to the supplemental material (Appendix Table S3. Abbreviations). The specific information is as follows:

Abbreviations	Full name
Scl-Ab	Sclerostin-neutralizing antibody
Glut1	Glucose transporter type 1
HK2	Hexokinase 2
PFK	Phosphofructokinase
GAPDH	Glyceraldehyde-3-phosphate dehydrogenase
PGK1	Phosphoglycerate kinase 1
PKM1/2	Pyruvate kinase muscle isozyme 1/2
LDHA	Lactate dehydrogenase A
PDK1	Pyruvate dehydrogenase kinase 1
PDH	Pyruvate dehydrogenase
HBP	Hexosamine biosynthetic pathway
O-GlcNAc	O-linked β -N-acetylglucosamine
OGT	O-GlcNAc transferase
OGA	O-GlcNAc hydrolase
GFAT1	Glutamine fructose-6-phosphate amidotransferase 1
PFKFB3	6-Phosphofructo-2-Kinase/Fructose-2,6-Biphosphatase 3
rhWnt3a	Recombinant human Wnt3a
DON	Diazooxonorleucine
TMG	Thiamet G

OSMI	O-GlcNAc transferase (OGT) small molecule inhibitor
Doxy	Doxycycline
BV/TV	Bone volume over tissue volume
Tb. N.	Trabecular bone number
Tb. Th.	Trabecular thickness
Tb. Sp.	Trabecular space
Ct. Th.	Cortical thickness
Tt. Ar.	Total area
Ct. Ar.	Cortical area
Ct. Ar./Tt. Ar.	Cortical area over total area
BMD	Bone mineral density
2-DG	2-Deoxy-D-glucose
ECAR	Extracellular acidification rate
OCR	Oxygen consumption rate
LiCl	Lithium chloride
β -cat	β -catenin
XAV939	Small molecule inhibitor of tankyrase (TNKS) 1 and 2
H89	Cyclic AMP-dependent protein kinase (protein kinase A) inhibitor H-89
STO	Calcium/calmodulin-dependent protein kinase kinase (CaMKK) inhibitor STO-609

- 3) The presentation of the surely important metabolite screening shown in Figure 1A should be modified, since the labeling is far too small to ensure readability of a printed version.

Response: Thanks for your suggestion. We have made changes in Figure 1A.

- 4) The experiment shown in Fig. S2 needs to be better explained in the Results section (the respective sentence is also incomplete).

Response: Thank you for your suggestions. We reintegrate Fig. S2 into Figure EV1 F, and the corresponding results are expressed in lines 168 to 172.

- 5) There is no specific information regarding the antibody that was used to inhibit sclerostin. Did you authors use a murine sclerostin antibody?

Response: Thanks for the kind reminder that we used a humanized sclerostin neutralizing antibody, specific information has been added in the mouse section of the methodology (line 504).

6) Regarding the mouse data presented in Figure 3/4 the authors have to be precise about the controls, as there are several possibilities, also in terms of doxycyclin removal. Most importantly, whereas it is first stated in the Results section that *Osx-Cre* mice were used as controls, the legend of Figure 3 refers to WT mice as controls. Moreover, while it is stated in the text that "the cancellous bone mass was still increased by Slc-AB in the *Ogt* CKO mice", this particular statistical comparison is not displayed in Figure 3C-E. For the sake of clarity, it might be an option to provide the respective p-values in the text. Finally, it is not informative to show the HE and Masson staining in Figure 3G/H and Figure 4E/F without quantification.

Response: Thanks for your suggestions. We are sorry for the confusion that we have caused. We used *Osx-Cre* as the Ctrl throughout this work, and the label of WT mice is a typo which has been corrected in this revision. We also added the p-value for each comparison in the figure to give better clarity in Figure 3C-E. Based on the comments of the reviewer, we also added the quantification of Masson staining in Figure EV2 B and Figure 4 H and I.

7) Whereas the molecular results are generally convincing and confirmed by quantitative analysis, that data regarding the 6-hour response towards *Wnt3a* are less convincingly presented. Especially since the "dramatically increased

intracellular calcium flux after 4 mins of rhWnt3a administration" shown in Figure 7E (red line) is not very evident, a quantification of the respective data should be provided. Moreover, there is no information about statistically significant differences regarding the Western blot results, and it might be more informative to provide this information by showing bar graphs. The same applies for the data shown in Figure 8B/E.

Response: Thank you for this informative suggestion. In Figure 7E, we have added the statistical analysis of the calcium fluxes of Veh and rhWnt3a after 260 s. There was about 2-fold change of Ca^{2+} flux in response to rhWnt3a compared to Veh which shown the Wnt ligand significantly increased the flux. In addition, bar graphs of the Western blot results of Figure 7 and Figure 8B and E have been supplemented to the Expanded View Figures (Fig. EV4 A to D and F to I, and Fig. EV 5 A to B).

8) Although it is surely important to utilize Wnt3a and ST2 cells in order to understand the molecular influence of the Wnt signaling pathway for osteogenesis, the authors should at least critically discuss that there is so far no human genetic evidence for a specific role of WNT3a in human bone mass acquisition (unlike it is the case for WNT1). Another important issue that should be discussed and/or experimentally addressed relates to the cell type specificity of the observed influences. In other words, does every cell type react to Wnt3a by induced O-GlcNacylation, or is this

specific for osteoblast progenitor cells?

Response: Thank you for this insightful suggestion. Whether *WNT3A* acts in the same way to promote osteogenesis in mice and humans has not been reported in relevant studies. Studies have shown that the *WNT3A* gene is strongly associated with bone fragility (Caetano da Silva *et al*, 2021; Velázquez-Cruz *et al*, 2014). Meanwhile, a study has shown that the *WNT3A* (c.152A > G, p.K51R) mutation is found in patients with Childhood-onset primary osteoporosis and that the mutation manifests in CHO cells as a reduction in the activity of the classical Wnt pathway, supporting the role of the classical Wnt pathway in the development of the disease (Korvala *et al*, 2012) (line 403-409). In this study, we found that O-GlcNAcylation was up-regulated after 72 hours of induction with rhWnt3a in C3H/10T1/2 and M2-10B4 cells (Fig. EV1 G and H), respectively, suggesting that up-regulation of O-GlcNAcylation of cellular total proteins by rhWnt3a may be a general phenomenon. Besides, the extent of expression of O-GlcNAcylation was not the same in different cells due to the treated period, concentration, and degree of cellular response to rhWnt3a treatment (line 398-403).

Reference:

Caetano da Silva C, Ricquebourg M, Orcel P, Fabre S, Funck-Brentano T, Cohen-Solal M, Collet C (2021) More severe phenotype of early-onset osteoporosis associated with

recessive form of LRP5 and combination with DKK1 or WNT3A. *Mol Genet Genomic Med* 9: e1681

Korvala J, Löija M, Mäkitie O, Sochett E, Jüppner H, Schnabel D, Mora S, Cole WG, Ala-Kokko L, Männikkö M (2012) Rare variations in WNT3A and DKK1 may predispose carriers to primary osteoporosis. *Eur J Med Genet* 55: 515-519

Velázquez-Cruz R, García-Ortiz H, Castillejos-López M, Quiterio M, Valdés-Flores M, Orozco L, Villarreal-Molina T, Salmerón J (2014) WNT3A gene polymorphisms are associated with bone mineral density variation in postmenopausal mestizo women of an urban Mexican population: findings of a pathway-based high-density single nucleotide screening. *Age (Dordr)* 36: 9635

Dear Prof. Shi,

Thank you for the submission of your revised manuscript to our editorial offices. I have now received the reports from the three referees that I asked to re-evaluate the study, you will find below. As you will see, the referees now fully support the publication of the study in EMBO reports.

Before we can proceed with formal acceptance, I have these editorial requests I ask you to address in a final revised manuscript:

- Please provide a final title with not more than 100 characters (including spaces).
- Please provide individual production quality figure files as .eps, .tif, .jpg (one file per figure), of main figures and EV figures (see below). Please upload these as separate, individual files upon re-submission.
- Please add up to 5 keywords to the manuscript and order the manuscript sections like this, using these names: Abstract - Keywords - Introduction - Results - Discussion - Methods - Data availability section - Acknowledgements - Disclosure and Competing Interests Statement - References - Figure legends - Expanded View Figure legends
- We now use CRediT to specify the contributions of each author in the journal submission system. CRediT replaces the author contribution section. Please use the free text box to provide more detailed descriptions and do NOT provide your final manuscript text file with an author contributions section. See also our guide to authors:
<https://www.embopress.org/page/journal/14693178/authorguide#authorshipguidelines>
- Please provide the Appendix file as .pdf and named 'Appendix'. Please start this file with a title page just stating "Appendix for ..." followed by the final title and then the table of contents. There is no need to repeat author names and affiliations.
- Please remove the list of abbreviations (Appendix Table S3) from the Appendix and define each abbreviation upon first mention in the manuscript text.
- Please move the 2 tables (siRNA and primer sequences) from the methods section to the Appendix. Please name these 'Appendix Table S3' and 'Appendix Table S4'. Finally, please make sure each Appendix table is called out in the manuscript text file.
- Please make sure that the number "n" for how many independent experiments were performed, their nature (biological versus technical replicates), the bars and error bars (e.g. SEM, SD) and the test used to calculate p-values is indicated in the respective figure legends. Please also check that all the p-values are explained in the legend, and that these fit to those shown in the figure. Please provide statistical testing where applicable. Please avoid the phrase 'independent experiment', but clearly state if these were biological or technical replicates. Please also indicate (e.g. with n.s.) if testing was performed, but the differences are not significant. In case n=2, please show the data as separate datapoints without error bars and statistics. See also:
<http://www.embopress.org/page/journal/14693178/authorguide#statisticalanalysis>

If n<5, please show single datapoints for diagrams. Moreover:

- Please note that the legend for figure 1h is mislabeled as figure 1g for the statistical information in the manuscript. This needs to be rectified.
- Please note that the exact p values are not provided in the legends of figures 1b, d-e, h; 2c, f, i, k, m; 3c, j, l-m, o; 4c-e, h-i; 5d-j; 6a-c, e-k, m, p; 7e; 8d, g, i; EV 1a; EV 2b, d, f; EV 3a; EV 4b-d, f-i; EV 5a-b. Please provide exact p-values.
- Please indicate the statistical test used for data analysis in the legends of figures 1a; 5a-b.
- Although 'n' is provided, please describe the nature of entity for 'n' in the legends of figures 4c-e, h-i; 6a-k, m, p; 8d, g, i; EV 4a-d, f-i; EV 5a-b.
- Please add to each legend (main, EV and Appendix figures, where applicable) a 'Data Information' section explaining the statistics used or providing information regarding replicates and scales. See:

- Please make sure that all figure panels are called out separately and sequentially. Presently, there seems to be no separate callout for panel 1H. Moreover, Fig. 9 is called out at the end of the results section, but there is no Fig. 9. Please check.
- Please add scale bars of similar style and thickness to microscopic images, using clearly visible black or white bars (depending on the background). Please place these in the lower right corner of the images themselves. Please do not write on or near the bars in the image but define the size in the respective figure legend. Presently, several scale bars have text nearby or are too thin. Please check.

- It seems the image showing the actin loading control in panel 5J and panel EV1D is identical. As these seem to show data from the same experiment, this is fine. However, please indicate in the respective figure legends that the same blot is shown in the panels and why.

- Thank you for providing the requested source data. Please upload this as one folder per figure (with all files for one figure in one folder and ZIPed).

In addition, I would need from you:

Please use this link to submit your revision: <https://embor.msubmit.net/cgi-bin/main.plex>

Best,

Referee #1:

The manuscript is suitable for publication in EMBO reports without further revision.

Referee #2:

The authors have adequately responded to my critiques. I do not have any further comments or concerns.

Referee #3:

The authors have adequately responded to my comments and further improved their manuscript.

All editorial and formatting issues were resolved by the authors.

Prof. Yu Shi
State Key Laboratory of Oral Diseases and National Center for Stomatology and National Clinical Research Center for Oral Diseases, West China Hospital of Stomatology, Sichuan University
China

Dear Prof. Shi,

I am very pleased to accept your manuscript for publication in the next available issue of EMBO reports. Thank you for your contribution to our journal.

Yours sincerely,
